# Pulsed electromagnetic field (PEMF) transiently stimulates the rate of mineralization in a 3-dimensional ring culture model of osteogenesis

**Paul D. Benya** [1], **Aaron Kavanaugh** [1], **Martin Zakarian**[1¤a], **Philip Söderlind**[2], **Tea Jashashvili**[3¤b], **Nianli Zhang**[4], **Erik I. Waldorff**[4], **James T. Ryaby**[4], **Fabrizio Billi**[1] *

**1** Department of Orthopaedic Surgery, David Geffen School of Medicine, University of California Los Angeles, Los Angeles, California, United States of America, **2** Department of Architecture and Urban Design, University of California Los Angeles, Los Angeles, California, United States of America, **3** Department of Radiology, Molecular Imaging Center, Keck School of Medicine, University of Southern California, Los Angeles, California, United States of America, **4** Orthofix Medical Inc., Lewisville, Texas, United States of America

¤a  Current address: UCLA AIDS Institute, School of Nursing, University of California, Los Angeles, Los Angeles, California, United States of America
¤b  Current address: Department of Integrative Anatomical Sciences, Keck School of Medicine, University of Southern California, Los Angeles, California, United States of America
* f.billi@ucla.edu

**Data Availability Statement:** Data in Dryad have been made publicly available at the following link: https://doi.org/10.5068/D17957.

## Abstract

Pulsed Electromagnetic Field (PEMF) has shown efficacy in bone repair and yet the optimum characteristics of this modality and its molecular mechanism remain unclear. To determine the effects of timing of PEMF treatment, we present a novel three-dimensional culture model of osteogenesis that demonstrates strong *de novo* generation of collagen and mineral matrix and exhibits stimulation by PEMF in multiple stages over 62 days of culture. Mouse postnatal day 2 calvarial pre-osteoblasts were cast within and around Teflon rings by polymerization of fibrinogen and cultured suspended without contact with tissue culture plastic. Ring constructs were exposed to PEMF for 4h/day for the entire culture (Daily), or just during Day1-Day10, Day11-Day 27, or Day28-Day63 and cultured without PEMF for the preceding or remaining days, and compared to no-PEMF controls. PEMF was conducted as HF Physio, 40.85 kHz frequency with a 67 ms burst period and an amplitude of 1.19 mT. Osteogenesis was kinetically monitored by repeated fluorescence measurements of continuously present Alizarin Red S (ARS) and periodically confirmed by micro-CT. PEMF treatment induced early-onset and statistically significant transient stimulation (~4-fold) of the mineralization rate when PEMF was applied Daily, or during D1-D10 and D11-D27. Stimulation was apparent but not significant between D28-D63 by ARS but was significant at D63 by micro-CT. PEMF also shifted the micro-CT density profiles to higher densities in each PEMF treatment group. Ring culture generated tissue with a mineral:matrix ratio of 2.0 by thermogravimetric analysis (80% of the calvaria control), and the deposited crystal structure was 50% hydroxyapatite by X-ray diffraction (63% of the calvaria and femur controls), independent of PEMF. These results were consistent with backscatter, secondary electron, and elemental

**Funding:** This research was supported by grants to FB from Orthofix Medical Inc, Lewisville, TX, USA; orthofix.com. PDB, AK, MZ, and FB, received salary support, and PS and TJ received fee-for-service support. NZ, EIW, and JTR are employed by Orthofix. Publication fees were paid by Orthofix. Orthofix participated in study design and preparation of the manuscript through the author contributions of NZ, EIW, and JTR. Support was also provided by grants to FB from the Orthopaedic Institute for Children, Los Angeles, CA, USA, OIC. com. FB received salary and general research support. OIC had no role in study design, data collection and analysis, decision to publish, or preparation of the manuscript.

**Competing interests:** The authors have read the journal's policy and the authors of this manuscript have the following competing interests: NZ, EIW, and JTR are employees of Orthofix, Inc. PDB, AK, and FB received salary support from Orthofix, Inc. TJ and PS received fee-for-service support from Orthofix, Inc. Publication fees were paid by Orthofix. PDB and FB received support to attend meetings and present data. FB received salary and general support from the Orthopaedic Institute for Children. Participation in this research by Orthofix and the Orthopaedic Institute for Children is described in the Financial Disclosure Statement. There are no patents, products in development, or marketed products to declare regarding the data in this publication related to the authors or either funding institution. Orthofix, Inc. markets medical devices that apply PEMF in clinical practice. This does not alter our adherence to PLOS ONE policies on sharing data and materials.

analysis by scanning electron microscopy. Thus, in a defined, strong osteogenic environment, PEMF applied at different times was capable of further stimulation of osteogenesis with the potential to enhance bone repair.

## Introduction

PEMF is currently in clinical use for enhancement of bone-related repair [1] based on efficacy in clinical trials in specific circumstances, namely, fracture non-union [2] and spinal fusion [3–5]. Clinical interest is motivated by the application of PEMF at a distance from the target site, the potential for mobile application, and its limited side effects. Despite this interest and efficacy, the technology is not yet fully exploited at the maximum of its potential and will benefit from optimization for the circumstances of the repair/prevention environment, the characteristics of the PEMF waveform, and the dose (duration and frequency of application). The incomplete understanding of the mechanisms involved and the lack of available, accurate *in vitro* models of osteogenesis for the elucidation of mechanisms central to the efficacy of PEMF greatly limits the application of PEMF. These constraints reflect the complexity of the molecular events [6–10] and the sequential nature of the process of bone formation [11–15]. The orchestrated differentiation of osteoblasts and then osteocytes, with attendant mineralization, depend on cell-cell, cell-matrix, and cell-growth factor interactions, as well as the absence of cell-plastic signaling. PEMF treatment of osteogenic precursors in a variety of culture models has been shown to stimulate cell proliferation [16–18] and induce expression of osteogenic markers [16–18]. Based on microarray analysis and supportive loss- and gain-of-function experiments, these responses appear to be mediated by enhanced expression of miRNA 21 and its suppression of Smad-7, an inhibitor of Smad-2 in the TGF-beta signaling pathway [18]. PEMF also stimulates the expression of L-type voltage-gated ion channels to alter intracellular calcium concentration with consequences for second messenger signaling [19]. In addition, PEMF increases the expression of the adenosine $A_{2a}$, and $A_3$ receptors, leading to increased proliferation, cAMP production, anti-inflammatory responses, and bone repair [20, 21]. Such studies are invaluable for applying PEMF, but direct molecular targets of PEMF and their signaling networks have not yet been identified. Success in this would allow efficient screening of PEMF characteristics for optimized effect in a variety of clinical situations and patient-specific therapies. However, it is important to distinguish between PEMF effects on deficits in the utilized culture models and PEMF effects on an intact and fully functioning osteogenic process. Here, we present a novel 3D ring culture model that exhibits strong osteogenesis based on the integrative outcome measure of mineralization. Further enhancement of osteogenesis is demonstrated under the influence of PEMF applied at different times and stages of this osteogenic culture. Thus, the ring culture model will be useful for further delineation of PEMF treatment protocols and mechanistic analyses, as well as investigations of other bone-related pathologies [1, 22–26].

## Materials and methods

### Cells

Timed-pregnant 022-CD-1 mice were obtained from Charles River Laboratories (Wilmington, MA 01887 US), delivered on E13, and individually housed in micro-isolator cages enriched with nestlet. Pre-osteoblasts were isolated from 160 postnatal day 2 mouse calvaria following

dissection of the parietal bones without sutures. Pups and mothers were sacrificed by isoflurane overdose, followed by decapitation of the pups using protocol #2014-009-13a approved by the UCLA Animal Research Committee. Calvariae were digested 4 times with trypsin (0.625%, Fisher Scientific) and bacterial collagenase P (2 mg/ml, Roche) mixed 1:1 immediately before each digestion for 15 min at 37°C in a 10 ml syringe with rotation at 10 rpm. Released cells were separated by filtration through a 70 μ nitex screen supported by porous polyethylene in the barrel of the digestion syringe before aspirating fresh enzymes into the syringe for the next digestion. Released cells were mixed with cold alpha-MEM containing 10% charcoal/dextran treated, heat-inactivated fetal bovine serum (FBS), and primocin (1/500, Invivogen). Pelleted cells were resuspended in the same medium and pooled from digests 2–4 before plating at 3–4 million cells/T75 flask. Cells were cultured for 4–5 days before ring construct formation.

## Preparation of rings, casting trays, and supports

Rings, casting trays, and ring supports were designed in Rhino 5 and manufactured with Surfcam 2017 using computer numerical controlled milling of 1/32", 3/16" and ¼" Teflon sheet, respectively The CAD files for the rings, ring supports, and casting trays can be found in the Dryad repository https://doi.org/10.5068/D17957 in the compressed zip archive "S1.Zip".

Rings were generated from 1/32" Teflon sheet (Ridout Plastics Co, San Diego, CA) texturized before machining by orthogonal scraping with 80 grit sandpaper held on the edge of a rectangular block. All Teflon parts were washed in ethanol, sonicated in acetone (4 times 30 sec at maximum power and 50% duty cycle) using a *Hielscher* (Mount Holly, NJ) probe sonicator, treated with 10% bleach for 30 min, washed 3 times with 70% ethanol and held in 70% ethanol at room temperature until use. Casting tray wells were pretreated for 24-72h at 4°C by filling just to the surface with 1% bovine serum albumin (BSA) in phosphate-buffered saline (PBS), primocin, and avoiding any wetting of the non-well surface. Rings were pretreated with 50% FBS in PBS, primocin, for at least 24h, then placed in freshly emptied casting wells, and the wells were refilled with BSA. These rings were used for construct generation after 72 h.

## Preparation of cellular ring constructs and cell culture

Cells were released with 0.05% trypsin, 0.48 mM EDTA; reactions were stopped with an equal volume of cold 10% FBS, alpha-MEM, primocin, and the cell pellets resuspended in room-temperature-citrated human plasma at $3 \times 10^6$ cells/ml. 96-well PCR plates were filled in alternating rows with 10 μl of 500 mM $CaCl_2$ in 50 mM 4-(2-hydroxyethyl)-1-piperazineethanesulfonic acid (Hepes), pH 7.5, and 10 μl of 10 U/ml of bovine thrombin (Sigma, Cat# T7513) and held at 4°C. One row of casting tray wells containing rings was aspirated and constructs generated by transferring 200 μl of cell suspension into the calcium well and then into the thrombin well before transferring to the ring in the casting tray. The plasma/cell suspension was delivered below, around, and above the ring without touching the casting tray surface outside the well. Filled casting trays with 18 rings were then incubated at 37°C for 1h to finish fibrin clot formation. Each ring construct was covered with PBS and then transferred with forceps to a 24-well plate containing a ring support and 2.0 ml of Differentiation Medium for culture in a $CO_2$ incubator.

Basal Medium consisted of phenol red-free alpha-MEM containing primocin and lot-tested charcoal/dextran-treated 10% FBS (Omega Scientific, Lot 166601). The FBS was additionally heat-inactivated by 5 min heating from 37 to 56°C, maintenance at 56°C for 30 min, and cooling in ice water before sterile filtration. Differentiation Medium was made fresh every feeding day by supplementing Basal Medium with -80°C concentrated stocks of the following components (final concentration): estradiol (10 nM) [27, 28], tranexamic acid, TXA (1mM) [29],

beta-glycerol phosphate (0.5 mM) [30], ascorbate (0.25 mM), and ascorbate-2-phosphate (0.40 mM). Differentiation Medium was fed Monday, Wednesday, Friday, and additionally supplemented with ascorbate-2-phosphate by direct addition to wells on Tuesday, Thursday, and Saturday to maintain a stable level of functional ascorbate [31]. TXA was removed from this medium on Day 9. Mineralization Medium was constructed the same as Differentiation Medium except it did not contain TXA but did contain Alizarin Red S (ARS) (0.5 mg/ml) [30] and 3mM beta-glycerol phosphate, to measure and support mineralization, respectively. Mineralization Medium was fed on the same schedule and started on Day 11.

ARS fluorescence was followed by repeated measurement of each well over the duration of the experiment in a BMG FLUOstar Omega plate reader using the well scanning protocol with Excitation at 544 nm and Emission at 590 nm. Gain was set at the beginning of the experiment based on previous values at termination so that detector saturation would not occur. Experimental data were corrected for fluorescence from parallel wells containing only supports and rings without cell constructs.

## PEMF treatment of cultures

The PEMF signal (HF Physio, 40.85 kHz frequency with 67 ms burst period) used in the current study is derived from an already FDA approved PEMF signal indicated for long-bone non-union fractures (PhysioStim™ from Orthofix Medical Inc.) [32]. The number of pulses, burst frequency, and signal amplitudes are also similar. Hence the pulse frequency and the signal characteristics are well within the range of what is being used in the clinical setting. This PEMF signal is used here to determine its potential benefit in an *in vitro* osteogenic environment and increase the diversity of PEMF parameters available for screening and optimization. The PEMF signal was applied to the cells for 4h/day based on the protocols in the text and Figures. Day 0 in this study was the time point when the cells were seeded in the ring culture system for 24 hours. Culture plates were exposed to PEMF [32] in a plexiglass shelved rack with electromagnetic Helmholtz coils above and below the cell culture shelves (S10 Fig). The 3-dimensional constructs from PEMF and no-PEMF control groups were cultured in identical $CO_2$ incubators, and the PEMF signal was checked periodically. Simulation of the distribution of EMF in the supporting rack was performed via dedicated software (Multiphysics Software, COMSOL, Burlington, MA). The EMF distribution was also verified with a Hall Effect transverse Gaussmeter probe (FW Bell 5186 Gauss Meter equipped with a SAD18-1904 axial probe, Berg Engineering, Rolling Meadows, IL). At the same time, the electromagnetic signal was evaluated by a digital oscilloscope (TBS1000C, Tektronix, Beaverton, OR). A Gaussmeter probe was used to measure the magnetic field intensity at three different positions on the rack, specifically, center, 8 cm from the center, and 1 cm from the edge. The distribution of magnetic field intensity in the rack allowed for the definition of a safe zone around the center where the EMF was deemed constant. The cell culture plates were placed in this area marked by white rectangular borders (S10 Fig). Both the simulation model and the probe verification demonstrated that the cells were exposed to a magnetic field varying from 1.14 to 1.24 mT depending on the proximity to the rack center, with 1.19 mT as the average.

## Collagen and mineral architecture-Scanning Electron Microscopy (SEM)

Ring constructs for visualizing collagen and mineral architecture were digested with rocking at 33˚C for 3h in Hepes buffered saline, pH 7.5 (HBS) containing 0.5% trypsin and 1% Triton X-100, pre-equilibrated to 33˚C. Rings were washed for 30 min at 33˚C with Hepes/Triton and then with HBS. Using a template, rings were cut into quarters. For collagen visualization, these were fixed for 2h with fresh 4% paraformaldehyde in PBS and demineralized by extraction

with 20 ml volumes of 20 mM EDTA, pH 7.5 at 33˚C overnight. Extractions were repeated twice over 72h, and the quarters were dehydrated with 35, 70, and 95% ethanol before storage at -20˚C. These mineral-free collagen samples were critical point dried in a Tousimis Auto-samdri-810 Critical Point Dryer (Rockville, Maryland) and then sputter-coated with 11 angstroms of iridium using an IBS-e sputter coater (Ted Pella, Redding, CA. Formerly: South Bay Technologies, San Clemente, CA) before imaging by SEM. Ring quarters for visualizing mineral architecture were not fixed but further processed by digestion at 37˚C for 18h in 5 ml of HBS containing 6M urea, 0.5 mg/ml proteinase K (20 mg/mL; Roche Applied Sciences, Indianapolis, IN) and 0.04% $NaN_3$. An additional aliquot of proteinase K was added, and the digestion was continued for 4h and terminated by addition of tris(2-carboxyethyl)phosphine to 58 mM and 30 min of continued incubation (modified from [33]). Quarters were washed three times with HBS and dehydrated with 35, 70 and 95% ethanol and stored -20˚C before sputter coating as above and SEM imaging.

## Micro-CT

ARS intravital stained ring constructs were washed twice for 15 min with PBS at room temperature and dehydrated with 35 and 70% ethanol before storage at -20˚C. Micro-CT was performed at the Molecular Imaging Center (MIC) of the University of Southern California (Los Angeles, CA) using a SCANCO Medical $\mu$CT 50 scanner with a voxel dimension of 0.005 mm. Polyethylene terephthalate glycol cylinders with indentations matching the ring tabs at their ends were used to hold the rings at the top of U50823 Scanco holders (14 mm diameter and 92 mm high). Holders were filled with 95% ethanol to keep the samples submerged in a uniform mineral-stabilizing environment during scanning. Parameters for micro-CT scanning were: energy: 55 kVp; current: 145 μA (8W); projections: 1000/180˚; filter: 0.5 mm aluminum (Al); integration time 500 ms; averaging data by 2. To calculate bone mineral density from x-ray images, a calibration phantom was included in each scan cycle at 55kVp, 0.5mm Al filter up to BH:1200 mg HA/cc. Image data was reconstructed as DICOM 16-bit signed image stacks. VGSTUDIO MAX version 3.3.2 software (Volume Graphics, Inc.; GmbH; Heidelberg, Germany) was used to segment grayscale voxels attributed to the Teflon ring from those of similar absorbance due to the mineralized matrix deposited in culture. Data were extracted from the resultant ring-free DICOM images through a custom MatLab (MathWorks, Natick, Massachusetts) program as per instructions from Scanco (Southeastern, PA).

## Thermogravimetric analysis and X-ray diffraction

Ring constructs cultured without ARS were terminated by fixing at 4˚C in 35 and 70% ethanol before storage at -20˚C. Tissue within the Teflon ring from construct quarters was then submitted to sequential thermogravimetric analysis (TGA) and X-ray diffraction (XRD). TGA was performed by Veritas Testing and Consulting (Denton, Tx) using a Mettler-Toledo TGA/DSC 3+ calibrated to traceable reference standards. The temperature program in S5 Fig was used and left an organic material-free mineral residue for subsequent high-resolution XRD [34].

Due to the small size of ring constructs and bone samples, ranging from 0.2 to 1 mm, the XRD diffraction data were measured on a Bruker AXS D8 Diffractometer equipped with an Apex11 CCD detector (an area detector) and a high-end Incoatec microfocus Cu X-ray source with multilayer optics. The samples were mounted on a glass fiber with a minimum of mineral oil and centered within the beam. Several frames were collected to complete the specified data set range. Intensity vs. 2θ scans were then obtained by integrating the data using Bruker software XRD[2] [35]. Average crystal size was calculated by comparing the profile width of a

standard profile (a Si substrate, measured under the same conditions as samples) with a sample profile according to the Scherrer formula. The sample FWHM (full width at half maximum) was obtained by profile fit of the peak at $2\theta = 26°$, which does not exhibit overlapping and is along the c-axis [34]. A built-in tool (provided by PANalytical X'Pert Highscore software) based on a relative intensity ratio method was used to determine the quantitative mineralogy of these samples. Standard sample phases from the Joint Committee on Powder Diffraction Standards database of Ca-Phosphate (reference ID 01-070-2065) and hydroxyapatite (reference ID 01-074-0566) were used to calculate the relative percent ratio of the two phases in the samples.

### SEM and elemental analysis

Ring quarters for SEM and elemental analysis were embedded in Optimal Cutting Temperature compound, flash-frozen in liquid nitrogen, and stored at -80°C. Frozen sections were collected on Cryofilm 2c (UConn Health Sciences, Molecular Core Facility), attached to 12.7 mm Zeiss aluminum pin subs (16111–9, Ted Pella, Redding, CA) using adhesive carbon dots (16084–1, Ted Pella, Redding, CA), and critical point dried before sputter coating with iridium, as above. All samples were imaged using a Zeiss Supra 40VP SEM (Carl Zeiss Microscopy, LLC, White Plains, NY). EDS was performed using an integrated Thermo Noran UltraDry System SIX EDS system (ThermoFisher, Waltham, MA). The details of SEM imaging are presented in the appropriate Figure legends.

### Statistics

Statistical comparisons were performed in Microsoft Excel (Redmond, Washington) augmented by the Real Statistics Resource Pack software for Excel (Release 6.8), Copyright (2013–2020) Charles Zaiontz. www.real-statistics.com. Data for the micro-CT analysis was extracted from DICOM images using a custom MATLAB program and subsequently transferred to Excel. $\alpha = 0.05$ was chosen as the threshold of statistical significance for all analyses. Statistical comparisons were made pairwise between the control group (No PEMF) and each of the other test groups (D1-D10, D11-D27, and D28-Term) for all comparisons to assess the effectiveness of PEMF at each time interval versus an absence of PEMF. Tukey HSD tests were chosen for all comparisons because these control for family-wise type 1 error across multiple pairwise comparisons. Error bars, for those figures which possess them (Figs 6A, 6B; 9 and 10, S1A and S1B Fig) indicate standard error. Error bands (Fig 7A–7D and S2A–S2D Fig) likewise indicate standard error. In Fig 7A–7D and S2A–S2D Fig, which display micro-CT total µg HA *vs*. HA mg/mL, data is smoothed with a 5x1 moving average filter.

## Results

### 3D ring culture

We have developed a 3-dimensional (3D) ring culture method to evaluate osteogenesis without interference by conventional cell-substrate interactions and in the absence of preexisting collagen matrix. Thus, only cellular activity controlled the deposition of collagen, mineral, and necessary bone-specific accessory proteins. Postnatal day-2 mouse calvarial pre-osteoblasts were suspended in human plasma and cast/polymerized within and surrounding a non-adhesive textured Teflon ring. Within 2 days, the constructs contracted to a 60-micron thick disk within and surrounding the ring. Tabs on the rings allowed the ring constructs to be suspended in medium without contact with tissue culture plastic and provided access to nutrients and oxygen from both sides (Fig 1). Culture in Differentiation Medium (alpha-MEM containing 10%

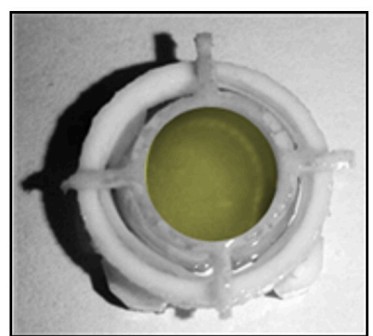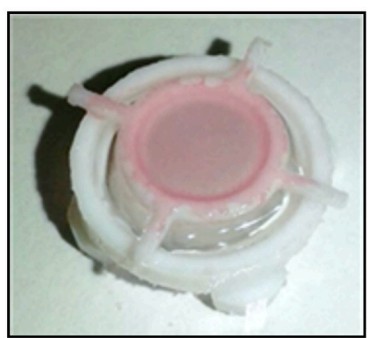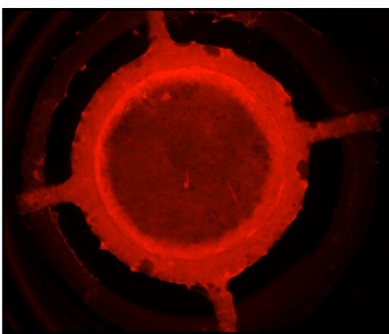

**Fig 1. Ring constructs at different stages of production/culture.** Left: Early culture in Differentiation Medium showing the Teflon ring and support (brightfield). Middle: Late culture after incubation in Mineralization Medium containing ARS and beta-glycerol phosphate (brightfield). Right: Late culture showing ARS fluorescence from a Daily PEMF-treated ring (exposure adjusted to reveal variation in intensity).

fetal bovine serum (FBS), 10 nM estradiol, ascorbic acid/ascorbate-2-phosphate, and TXA) with later addition of 3 mM beta glycerol phosphate and ARS (Mineralization Medium) induced robust osteogenesis demonstrated by distinct cellular production of alkaline phosphatase activity (a cellular marker of osteoblasts in this context) at both the surfaces and the interior of the construct, and by the deposition of a core of strongly mineralized matrix, as measured by intravital ARS staining (S9 Fig). At termination of culture, Day 63, extensive collagen deposition and mineralization of constructs were documented by SEM analysis after exposure to proteolytic removal of non-collagenous proteins, lipid and mineral (collagen visualization), or removal of all protein and lipid (mineral visualization) (Fig 2A and 2B,

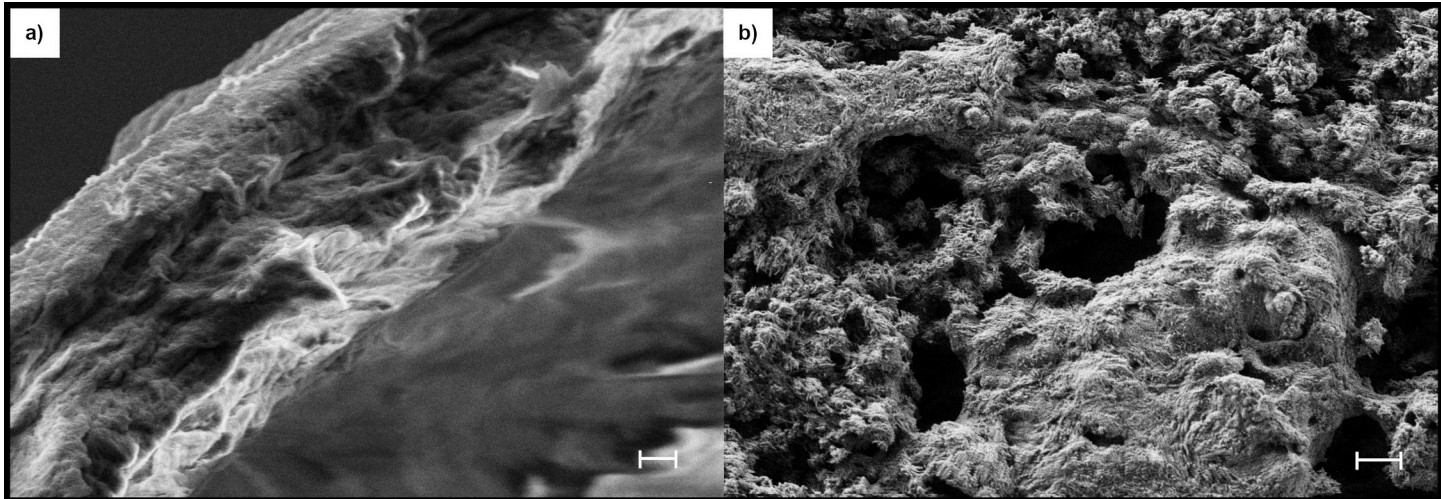

**Fig 2. Collagen and mineral deposited during ring culture.** Constructs at termination (Day 63) were differentially processed (see Methods) to expose just collagen (a) or just mineral (b) and visualized by SEM. Cut edges of the ring constructs are presented to allow visualization of the constructs' interior and part of the surface. SEM settings for (a): EHT = 5 KeV, WD = 4.7 mm, H = 30 μm, W = 40 μm, Mag = 2.96 K X. Scale Bar = 2 μm. SEM settings for (b): EHT = 3 KeV, WD = 2.1 mm, H = 122.7 μm, W = 163.6 μm, Mag = 699 X. Scale Bar = 10 μm.

respectively). Additional characterization is presented below, along with an evaluation of the effects of PEMF on this process.

## Effect of PEMF on mineralization

*In vitro* osteogenesis proceeds through sequential phases of osteoblast differentiation (with dependent collagen deposition), selective osteocyte differentiation (with dependent mineralization), and further deposition and maturation of the mineralized matrix [13]. We have used mineralization as a kinetic integrative measure of osteogenesis in evaluating PEMF because it is dependent on both osteoblast and osteocyte differentiation. Trace fluorescence labeling with the mineral-specific dye ARS as an intravital dye during culture both localizes and quantifies mineral deposition. We have used such labeling to non-destructively monitor/quantify this process in control and PEMF-treated cultures. Importantly, ARS fluorescence has been shown to localize and coincide with the appearance of osteocytes genetically tagged with an osteocyte-specific *Dmp-1* fluorescent reporter [30]. Thus, mineral-dependent ARS fluorescence in the present study documents progression along the osteogenetic lineage and marks osteocyte differentiation.

The overall experimental design to evaluate the effects of PEMF is presented in Fig 3, where PEMF was applied during three periods of culture corresponding to osteoblast differentiation, Day 1-Day 10 (Protocol A); osteocyte differentiation, Day 11- Day 27 (Protocol B); and continuing mineralization, Day 28-Day 63 (Protocol C). These periods of PEMF treatment were preceded and/or followed by culture in the absence of PEMF and assessed along with Daily PEMF (Protocol D, corresponding to current clinical use) by comparison to control culture, which received no PEMF during the course of the experiment. This permitted determination of

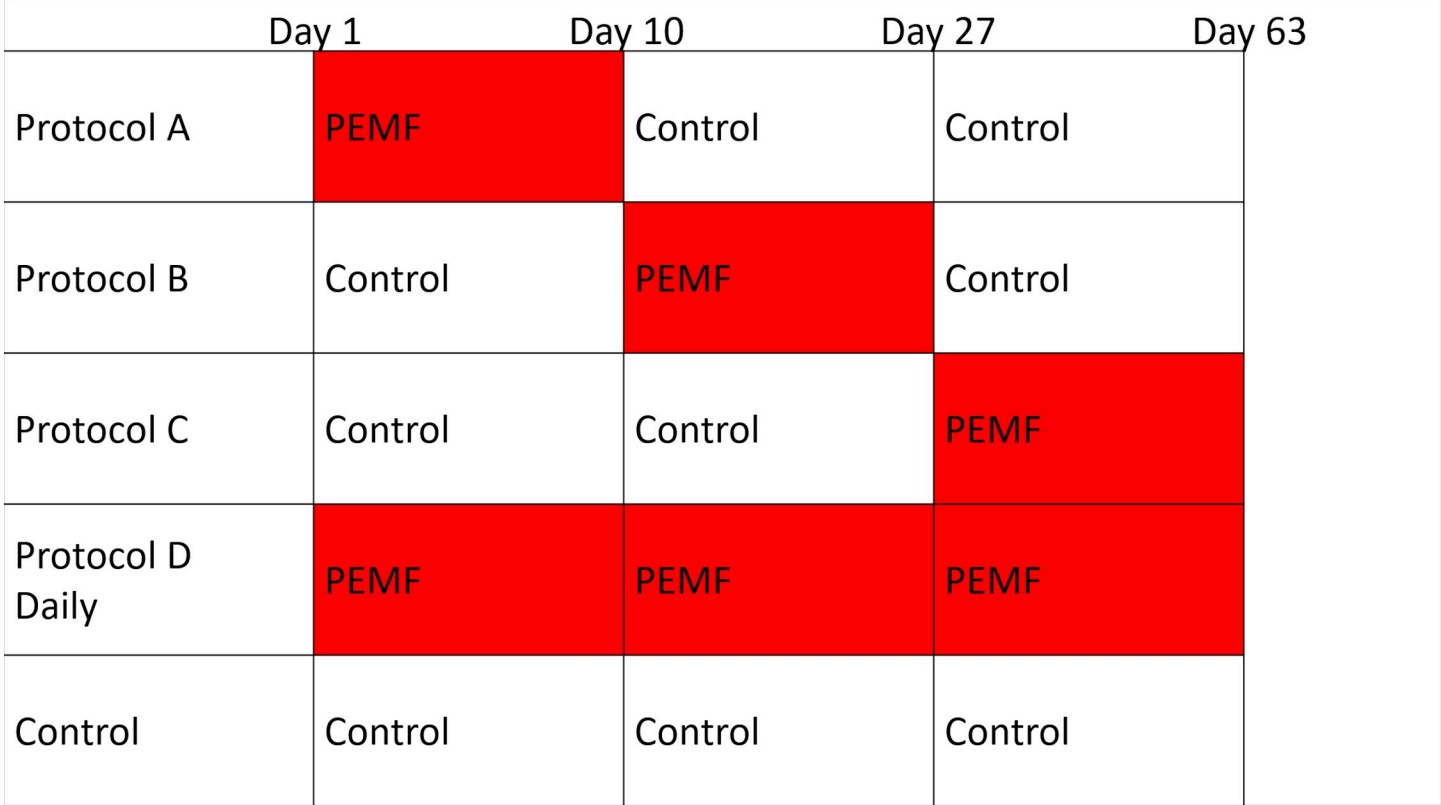

**Fig 3. Protocols for application of PEMF during ring culture.**

whether PEMF selectively enhanced any stage of osteogenesis. In these cultures, PEMF was defined as HF Physio, 40.85 kHz frequency with 67 ms burst period [36], and applied for 4h/day.

When PEMF was applied during the first 10 days of culture (Protocol A) a transient stimulation of mineralization was induced after Day 10, when Mineralization Medium with 3 mM beta-glycerol phosphate and ARS was first introduced (Fig 4A). Thus, the effect of PEMF was demonstrated even after PEMF treatment of cultures was discontinued. The stimulation was caused by the rapid onset of mineralization in PEMF cultures; ARS fluorescence was 3.9-fold greater in Day 16 PEMF cultures compared to control, no-PEMF cultures. Between Day 16 and 23, the rates of mineralization were similar and highly statistically significant in both types of cultures, as indicated by the parallel lines separated by the early mineralization caused by PEMF (Fig 4A). Subsequently, mineralization increased but converged to a common value for both PEMF and no-PEMF culture. When PEMF was introduced at Day 11 (Protocol B) (Fig 4B), it induced a similar rapid onset, 3.6-fold at Day 16. Subsequently, the rates were similar but converged with controls more slowly than when PEMF was introduced at Day 1.

PEMF treatment begun at Day 28 (Protocol C) also stimulated mineralization, but the increase in rate was gradual, detected by a trend in means (Fig 4C). The lack of statistical significance was due to the smaller number of replicates due to sampling later in culture and a

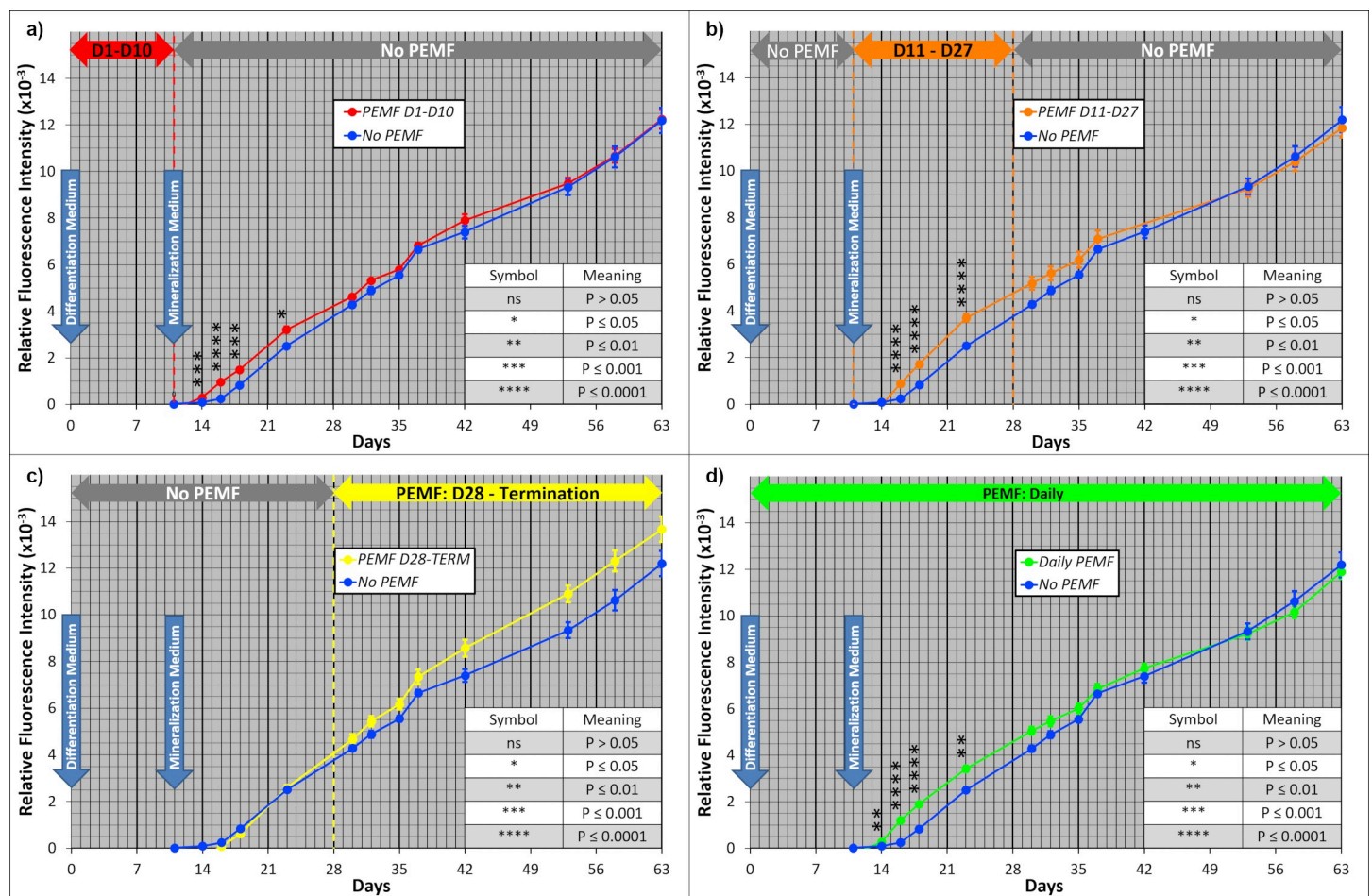

**Fig 4. Effect of PEMF on mineralization measured by intravital ARS fluorescence.** PEMF was applied to ring cultures according to Protocols A, B, C, and D, and the resultant ARS fluorescence is presented in panels (a), (b), (c), and (d); in each case compared to no-PEMF control cultures. In each case, n = 12 between Day 1 and Day 10; n = 9 between Day 11-Day and Day 27; and n = 6 between Day 28 and Day 63. Standard Error bars are presented with significance indicated by asterisks in the legend.

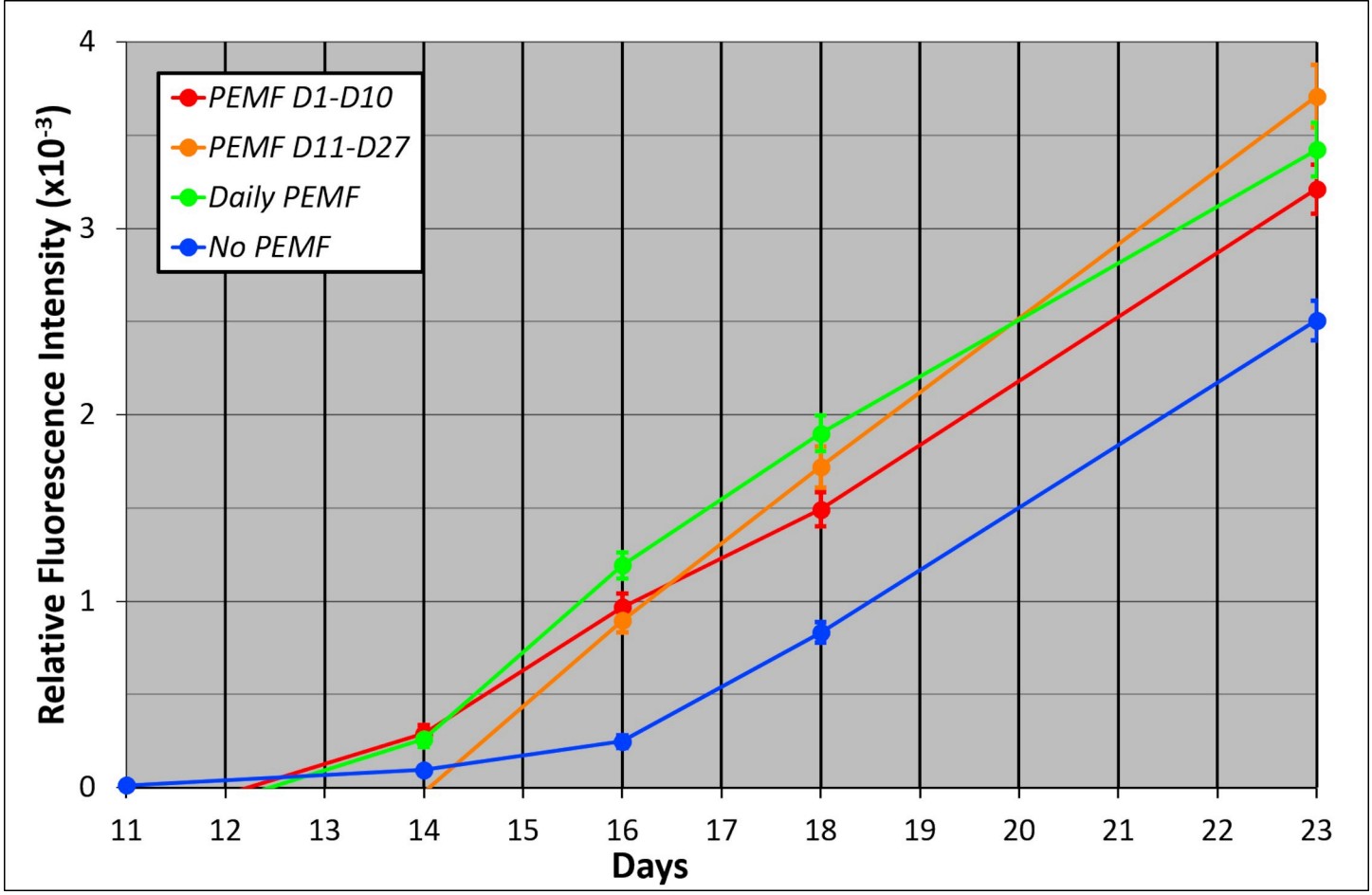

**Fig 5. Comparison of early PEMF-induced ARS mineralization responses for Protocol A, B, and D.** Data have been combined from Fig 4A, 4B and 4D in this emphasized narrow time frame. P-values for comparison to no-PEMF controls were uniformly less than 0.01 and can be found in Fig 4 but have been removed here for visual clarity. Fold increase compared to no-PEMF controls at Day 16 were 3.91, 3.61, and 4.81 for PEMF at D1-D10, D11-27, and Daily, respectively.

tendency toward greater data variation during this phase. The onset and rate of mineralization prior to PEMF treatment on Day 28 in Protocol C was identical with the control group, validating the system. In contrast, Daily PEMF treatment from the outset of culture (Fig 4D) yielded a mineralization profile essentially identical to that obtained from PEMF treatment from Day 1-Day 10; i.e., 4.8-fold PEMF enhancement on Day 16, separation by 900 ARS fluorescence units, and convergence by Day 37. For clarity, the early PEMF-induced responses are compared in Fig 5, and demonstrate the highly parallel and robust stimulation of mineralization produced by all 3 PEMF protocols (Protocols A, B, and D) that were active during this period of culture. After early PEMF exposure and stimulation, subsequent exposure had little effect. Importantly, the potential for a PEMF-responsive process is stable during the first 28 days of culture encompassing osteoblast and osteocyte differentiation, and can then be activated by PEMF, but with a slower time-course (Protocol C).

Micro-CT was used to extend the characterization of mineralization and support the results obtained by ARS fluorescence. At termination (Day 63) with high levels of mineralization, intravital ARS fluorescence did not discriminate between the different PEMF experimental groups and no-PEMF controls (S1A Fig). However, micro-CT identified a statistically significant elevation of mineral content compared to controls for both PEMF treatment between

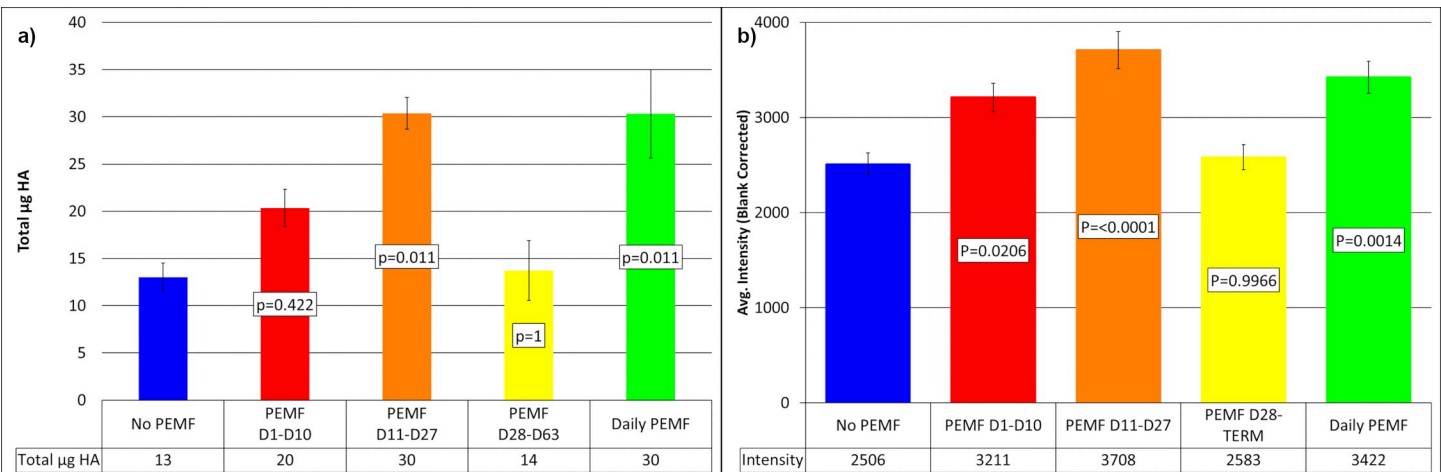

**Fig 6. Mineralization measured by micro-CT and ARS early in culture.** (a) Micro-CT of samples at Day 27 of culture. Standard error with n = 3 per group. (b) ARS fluorescence of samples at Day 23. Standard error with P = Tukey HSD, n = 9. Both analysis techniques yielded similar stimulation of mineralization by PEMF. Differences between groups are dependent on the time of analysis relative to the onset of PEMF. Day 28-Day 63 samples have not yet been exposed to PEMF and thus match no-PEMF controls.

Day 28 and 63, as well as for Daily PEMF (S1B Fig). In contrast, both micro-CT (Fig 6A, Day 27) and ARS (Fig 6B, Day 23) generated similar profiles earlier in culture in which differences between PEMF treatments and controls were readily apparent. At these times, the Protocol C group with treatment scheduled for Days 28 to 63 had not yet been exposed to PEMF, and thus, was comparable to the no-PEMF control. Consequently, the different detection mechanisms of ARS and micro-CT produced equivalent information and demonstrated significant effects of PEMF on biomineralization.

Greater detail concerning the enhancement of mineralization by PEMF was obtained from micro-CT by plotting the mass of hydroxyapatite (HA) equivalents at each mineral density (µg HA/ml). To visualize the effect of PEMF, the data from controls were subtracted from those from PEMF-treated constructs to create difference plots (Fig 7). In each case (PEMF on Day1-Day 10, Day 11-Day 27, and Daily) (Fig 7A, 7B and 7D, respectively), PEMF induced not only more mineral deposition but also shifted this mineral to higher mineral densities. The lone exception was PEMF during Day 28-Day 63, a group not yet exposed to PEMF at this time, and thus a useful no-PEMF control for the accuracy of the difference plots (Fig 7C). The increased HA mass and shift in mineral density persisted at termination (S2A–S2D Fig). This included the samples treated with PEMF between Day 28–63, supporting the capacity of PEMF to increase mineral mass and density during the later stages of culture (S2C Fig). Maximal PEMF-induced increase in mineral density at termination was detected in the range of the density from 200 to 225 HA mg/ml. When just these voxels were displayed in 3D and compared to those from controls, the PEMF voxels showed increased frequency but no apparent change in distribution throughout the ring construct (Fig 8).

Micro-CT was also used to compare the constructs at termination (Day 63, approximately 2 months old) with a calvaria from a 6-month-old mouse (S3 Fig). The density profiles demonstrated that neither the control nor PEMF-treated constructs matched that of the older calvaria, suggesting a requirement for more matrix maturation and/or exposure to *in vivo* mechanical or hormonal factors. Such incomplete development of mineralized matrix was also apparent when representative micro-CT cross-sections were compared to those of calvaria (S4 Fig). However, the geometry and uniformity of the construct mineral, with or without PEMF,

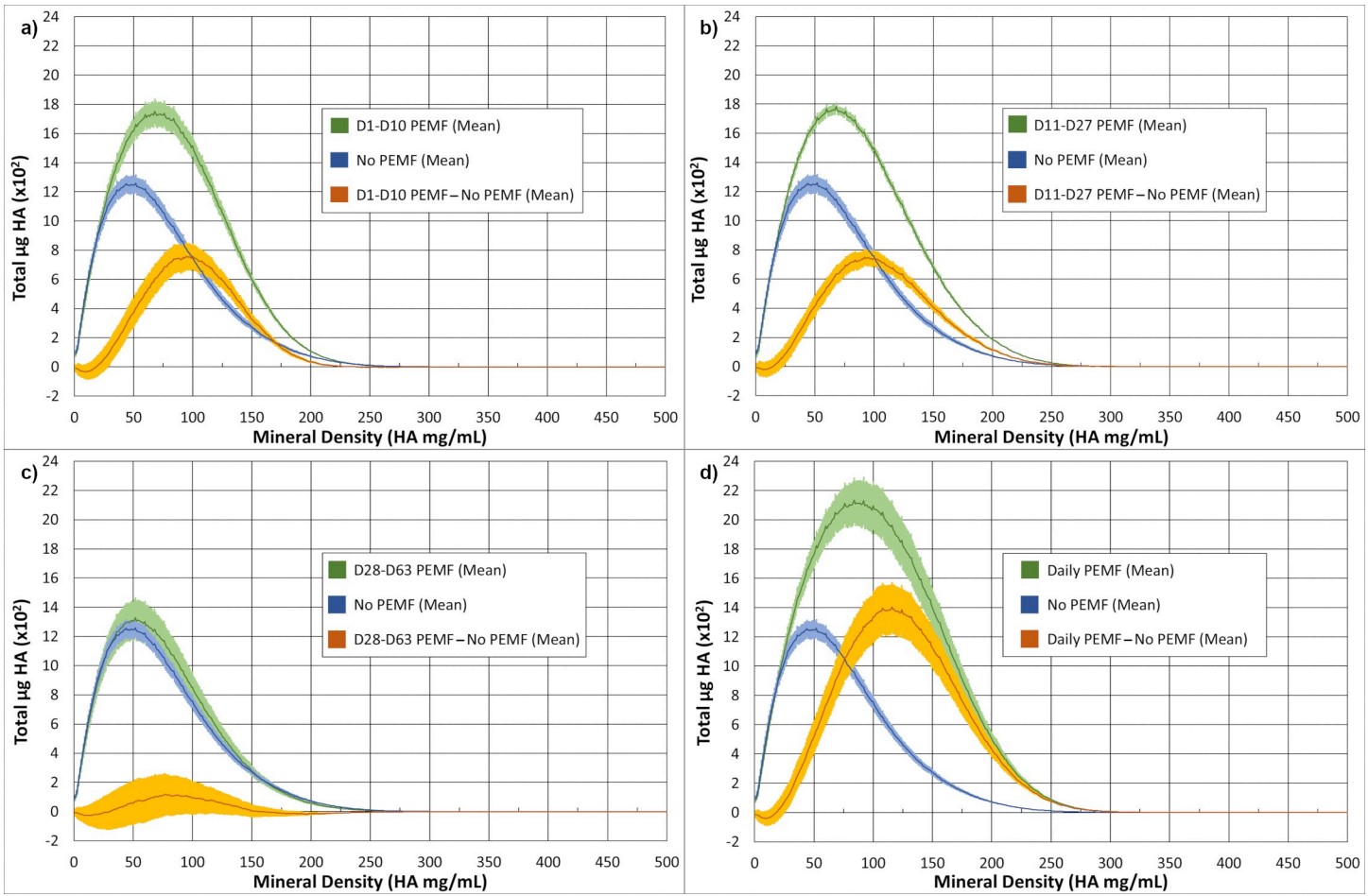

**Fig 7. Micro-CT-derived density profiles at Day 27.** Total HA vs. HA mg/ml was plotted, and profiles compared between no-PEMF controls and PEMF treatment between Day 1- Day 10 (a); PEMF between Day 11-Day 27 (b); PEMF between Day 28-Day 63 (c); and Daily PEMF (d). In each case, n = 3 and the lines are the means, and thicknesses represent the standard error. Control values were also subtracted from PEMF values at each density to yield a difference plot (orange) to visualize the effect of PEMF on mineral density.

was evident. In addition, it was apparent that the lower densities in the micro-patterned calvaria mineral were similar to the density of the constructs, suggesting their approach to the character of *in vivo* bone.

## Thermogravimetric analysis

Thermogravimetric analysis (TGA) was used to determine the bulk mineral/matrix ratio in the ring constructs compared to calvaria and femur. Tissue constructs were removed from the interior of the ring and submitted to controlled temperature gradients and continuous weight measurements (S5 Fig). Dry weights were determined after incubation at 200˚C, mineral weight, after incubation at 600˚C, and matrix weight was calculated as the difference. In bone, the majority of matrix weight is derived from collagen. It plays both structural and regulatory roles, as well as serves as a nucleation site for the onset of mineralization and subsequent accumulation of mineral. The mineral/matrix ratio of PEMF-treated constructs did not differ from that of controls (Fig 9). This result allows for PEMF-induced increases in mineral content/density if similar increases in collagen content occur in parallel. The *in vitro* deposited construct from controls yielded a mineral/matrix ratio of 2.0. This was 80% of the 2.5 ratio obtained for

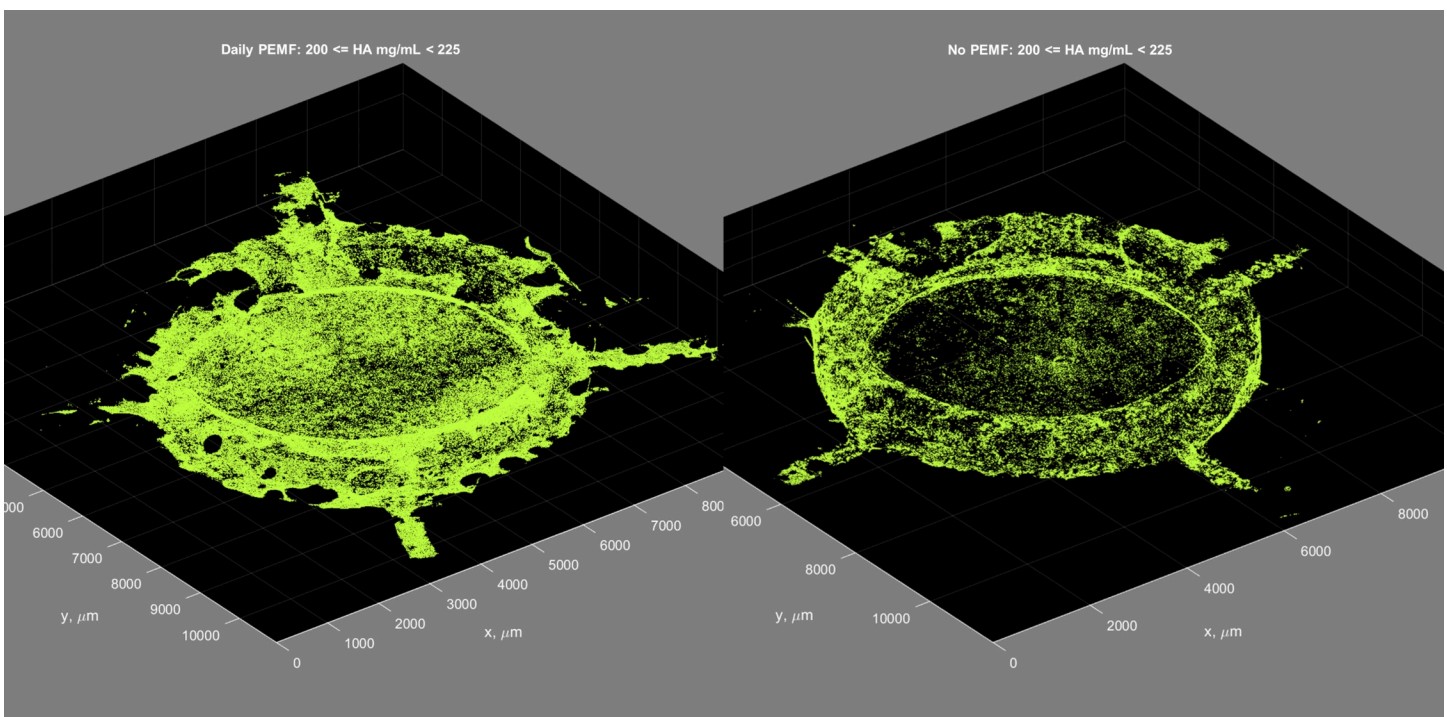

**Fig 8. Micro-CT-derived 3D distribution of voxels in the density range maximally influenced by PEMF on Day 63.** Constructs treated with Daily PEMF (left panel) and no-PEMF controls (right panel). The density range presented is 200–225 HA mg/ml.

calvaria and suggests that the ring constructs were approaching the bone's composition *in vivo*. Expectedly, the ratio of the femur was 3.6, consistent with its greater cortical bone density.

## X-ray diffraction determination of hydroxyapatite content

X-ray diffraction (XRD) was used to determine the content of hydroxyapatite (HA), the predominant calcium-phosphate crystal structure in the bone. The samples after TGA analysis were used for XRD because the temperature-dependent combustion of the organic matrix removed complicating/irrelevant reflections to yield defined, interpretable profiles using a single crystal adaptor. In addition, exposure to 800˚C heat drives the conversion of some non-HA crystals to tricalcium phosphate and degrades carbonate substituted HA, the latter a marker of bone aging [34]. The XRD profile for the femur is presented in S6 Fig and demonstrated 79% HA, 29% tricalcium phosphate, and mean crystallite size of 29.3 nm. The profile for a no-PEMF construct yielded 47% HA, 53% tricalcium phosphate, and a mean crystallite size of 16.4 nm, while the profile for PEMF treatment between Day 28-Day 63 yielded 48% HA, 52% tricalcium phosphate, and a mean crystallite size of 16.8 nm (S7 Fig, top and bottom, respectively). The HA content in all PEMF-treated constructs varied around 50% and did not differ from no-PEMF controls (Fig 10), while calvaria and femur contained 80% HA. Thus, PEMF did not influence crystal structure even though it increased mineral density and mineral content.

## SEM analysis of mineral content

Frozen sections of ring cultures at Day 63 were made to enable morphological and elemental analysis of cross-sections by SEM and EDS. High-resolution secondary electron images

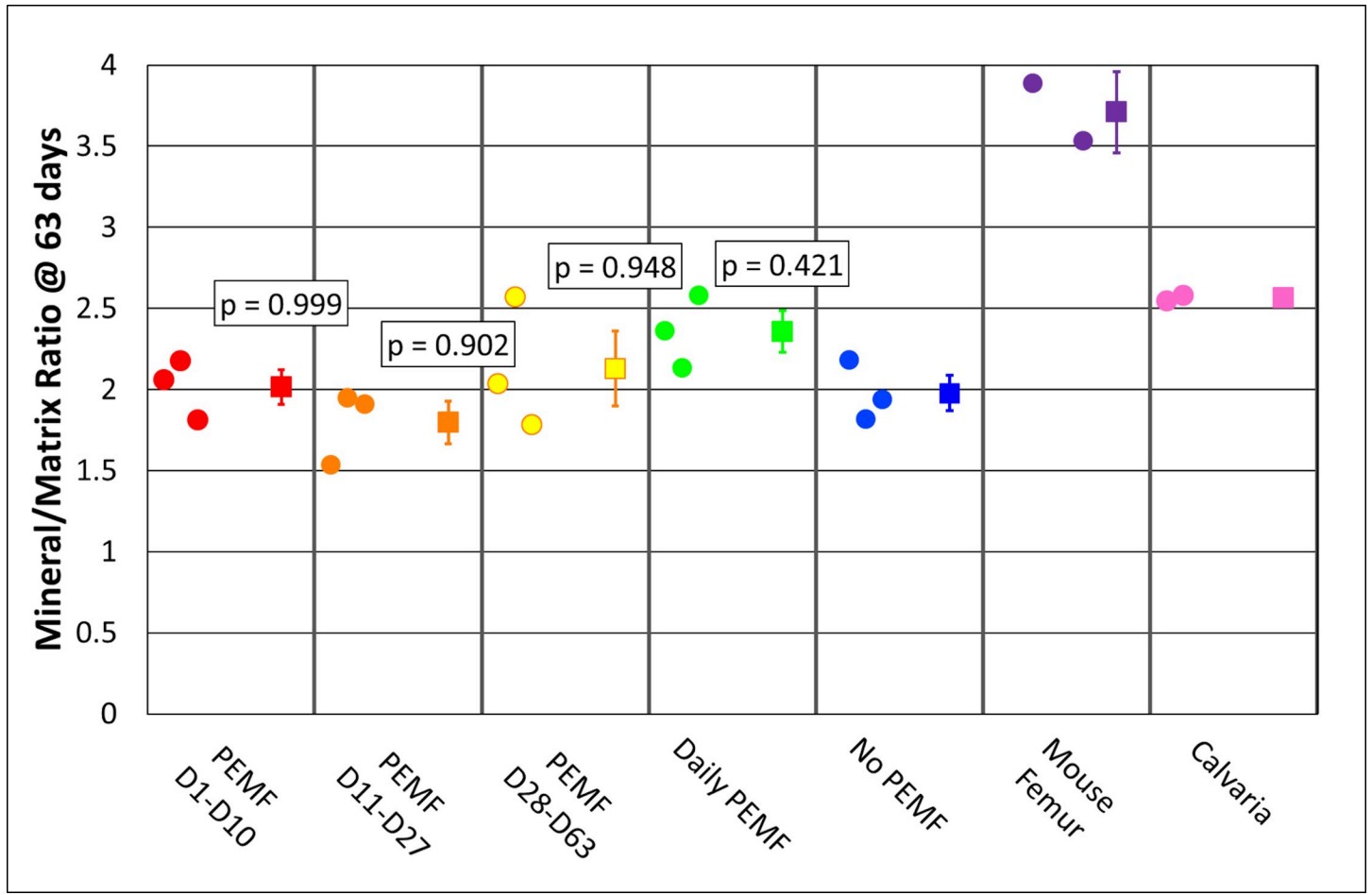

**Fig 9. TGA analysis for mineral/matrix ratio of ring constructs on Day 63, comparison with calvaria and femur.** Each group column presents individual values on the left and the mean and standard error on the right if n is 3 or greater. p-value = Tukey HSD, alpha = 0.05.

defined the limit of the construct and demonstrated packed deposited mineralization domains (Figs 11A and 12A) that reflected the collagen and inorganic mineral shown in Fig 2A and 2B, respectively. In addition to emphasizing these domains at the cut surface of the section, the images also demonstrated a smooth top and bottom surface on the constructs. Images obtained from backscattered electrons (Figs 11B and 12B) exhibited greater contrast of the mineralization domains due to the increased intensity obtained from elements with higher atomic numbers (Ca and P). In addition, these images document thin mineral-free zones at the top and bottom construct surfaces that appeared as shadows where the matrix was evident in secondary electron images. These zones may be equivalent to osteoid composed of a fine matrix of collagen fibrils (S8 Fig) deposited by osteoblasts that do not deposit mineral due to environmental cues. This conclusion was also supported by the distribution of cells, mineral and alkaline phosphatase in frozen sections when visualized by fluorescence microscopy (S9 Fig). The composition of the mineralization domains and the distribution of Ca and P were determined on the same sections in Figs 11B and 12B with EDS mapping (Figs 13 and 14). The separate maps for Ca and P both demonstrated mineralization domains that covered the interior of the constructs and were entirely overlapping. Ca/P ratios of 1.46 and 1.29 for PEMF and no-PEMF samples were consistent with a mixture of hydroxyapatite, calcium-deficient

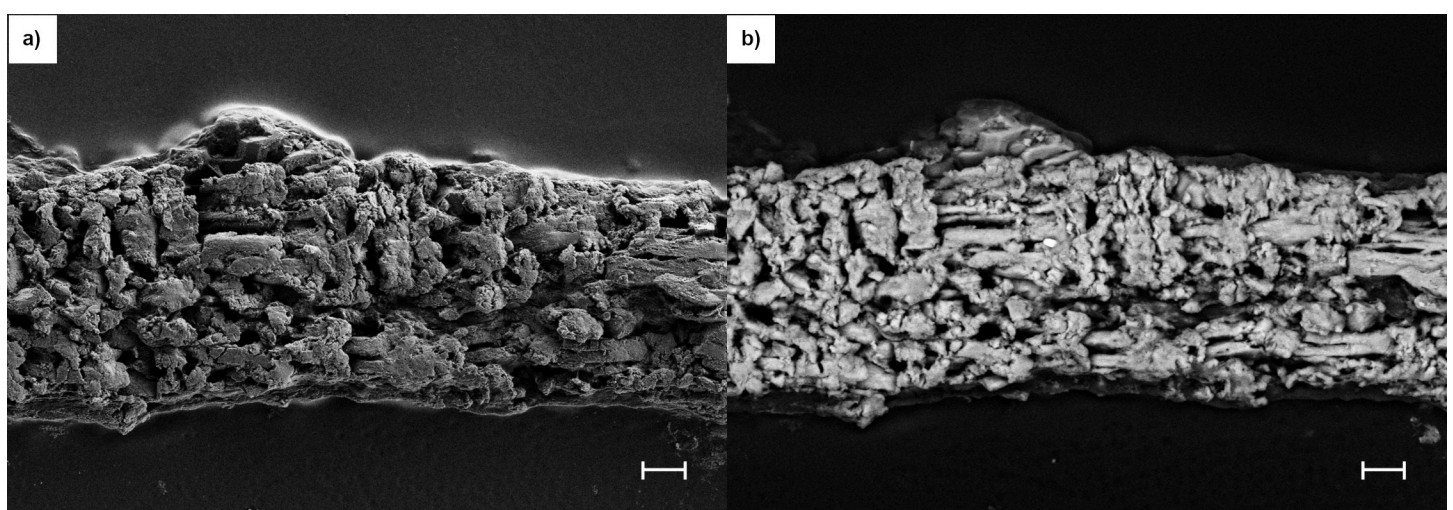

**Fig 10. XRD analysis of HA content in PEMF-treated and control samples compared to calvaria and femur.** Each group column presents individual values on the left, and the mean and standard error on the right if n is 3 or greater, *p*-value = Tukey HSD, alpha = 0.05.

**Fig 11.** SEM secondary (a) and backscattered (b) electron images from a Daily PEMF ring construct frozen section at Day 63. SEM settings for (a): EHT = 3KeV, WD = 4.7 mm, H = 128.3 μm, W = 171.0 μm. Scale Bar = 10 μm. SEM settings for (b): EHT = 15KeV, WD = 10.7 mm, H = 128.3 μm, W = 171.0 μm. Scale Bar = 10 μm.

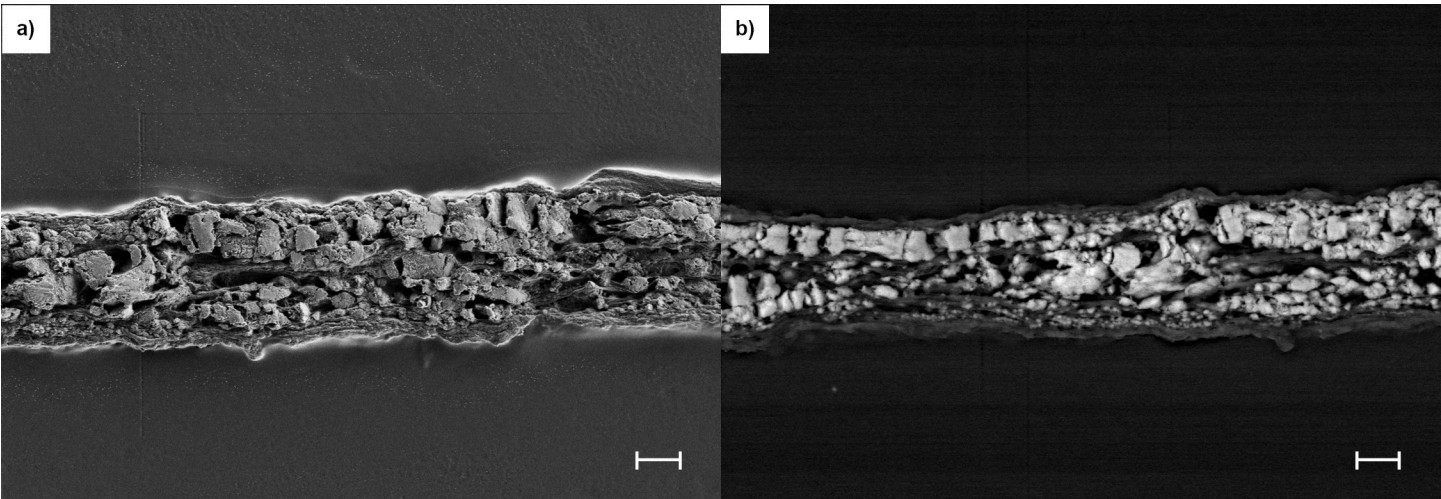

**Fig 12.** SEM secondary (a) and backscattered (b) electron images from a no-PEMF ring construct frozen section at Day 63. SEM settings for (a): EHT = 3KeV, WD = 4.4 mm, H = 128.3 μm, W = 171.0 μm. Scale Bar = 10 μm. SEM setting for (b): EHT = 15KeV, WD = 11.0 mm, H = 128.3 μm, W = 171.0 μm. Scale Bar = 10 μm.

hydroxyapatite, and carbonated hydroxyapatite, and tricalcium phosphate, and within the range reported for 1- and 6-week-old postnatal calvaria [37]. Hydroxyapatite content from XRD (Fig 10) supported this conclusion.

## Discussion

### Ring culture

The 3D osteogenic ring culture system presented here facilitates the evaluation of PEMF as a stimulator of osteogenesis. This system provides extensive *de novo* deposition of collagen and mineral, the two main structural components of bone. It is important that this process proceeds as a fully functioning system so that increases in osteogenesis by PEMF are not just the result of correcting a culture model but represent stimulation of a process that is similar to bone development/repair. To this end, the attachment-independent 3D culture environment, stable ascorbate protocol [31], mediating effects of estradiol (S9 Fig) [27, 28, 38], and the robust *de novo* deposition of prerequisite collagen matrix provide the basic environment for vigorous differentiation of osteoblasts and osteocytes and attendant mineralization. We have focused on mineralization as a late outcome measure of osteogenesis because, in a naïve environment, it requires strong cellular differentiation and activity and thus integrates the success of these steps. This conclusion is supported by the work of Dallas [30] using a highly similar calvaria pre-osteoblast, but monolayer, culture model of osteogenesis with pre-osteoblasts that express GFP under the control of the DMP-1 promoter. Thus, the expression of GFP in this system was a surrogate of DMP-1 expression, an acknowledged molecular marker of osteocytes. When GFP and ARS mineral fluorescence were simultaneously measured during osteogenesis in culture by dual-channel time-lapse fluorescent microscopy, these signals were parallel in induction and intensity and colocalized. This demonstrates that only cells expressing an osteocyte phenotype were responsible for mineral deposition, not fibroblasts or undifferentiated cells.

Similarly, *in vitro* osteogenesis of the clonal cell line, IDG-SW3 produced parallel ARS-detected mineral deposition and DMP-1 expression with colocalization. This pattern closely resembled the expression of other acknowledged early osteocytic markers E11, MEPE, PHEX,

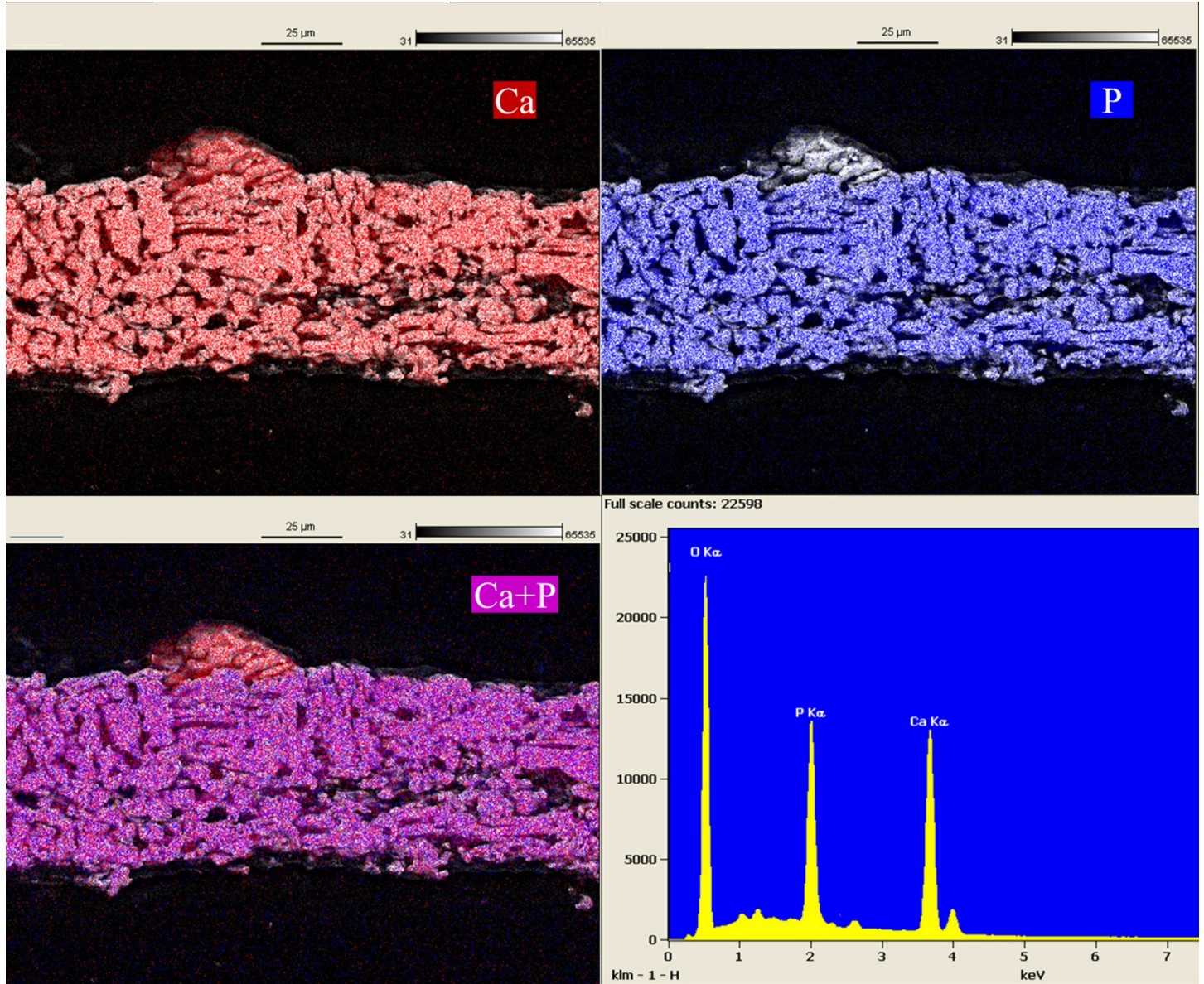

**Fig 13. SEM-EDS maps and spectrum from the same section of a Daily PEMF ring construct in Fig 11 with emphasis on Ca, P localization, and their overlay.** Ca/P ratio = 1.46. SEM settings: EHT = 15KeV, WD = 10.7 mm, H = 128.3 μm, W = 171.0 μm.

and the early osteoblast marker, alkaline phosphatase [13]. Consequently, mineralization measures these cellular differentiation processes as well as reflects the functional consequence of mineralized matrix deposition essential for bone.

Developing and maturing bone are frequently characterized by the mineral/matrix ratio (whether in bulk by TGA [34], or locally by Raman microspectroscopy [39, 40] or FTIR [37]), and this parameter has been shown to continue to increase between 2 weeks and 6 months of age in mice [37, 39]. At the end of ring culture (only two months), the experimental mineral/matrix ratio had already reached 80% of the 6-months-old calvarial control that was processed and analyzed in parallel with our experimental samples. Given the time-dependent trajectory of the mineral-dependent ARS fluorescence and the increase in micro-CT-determined mineral

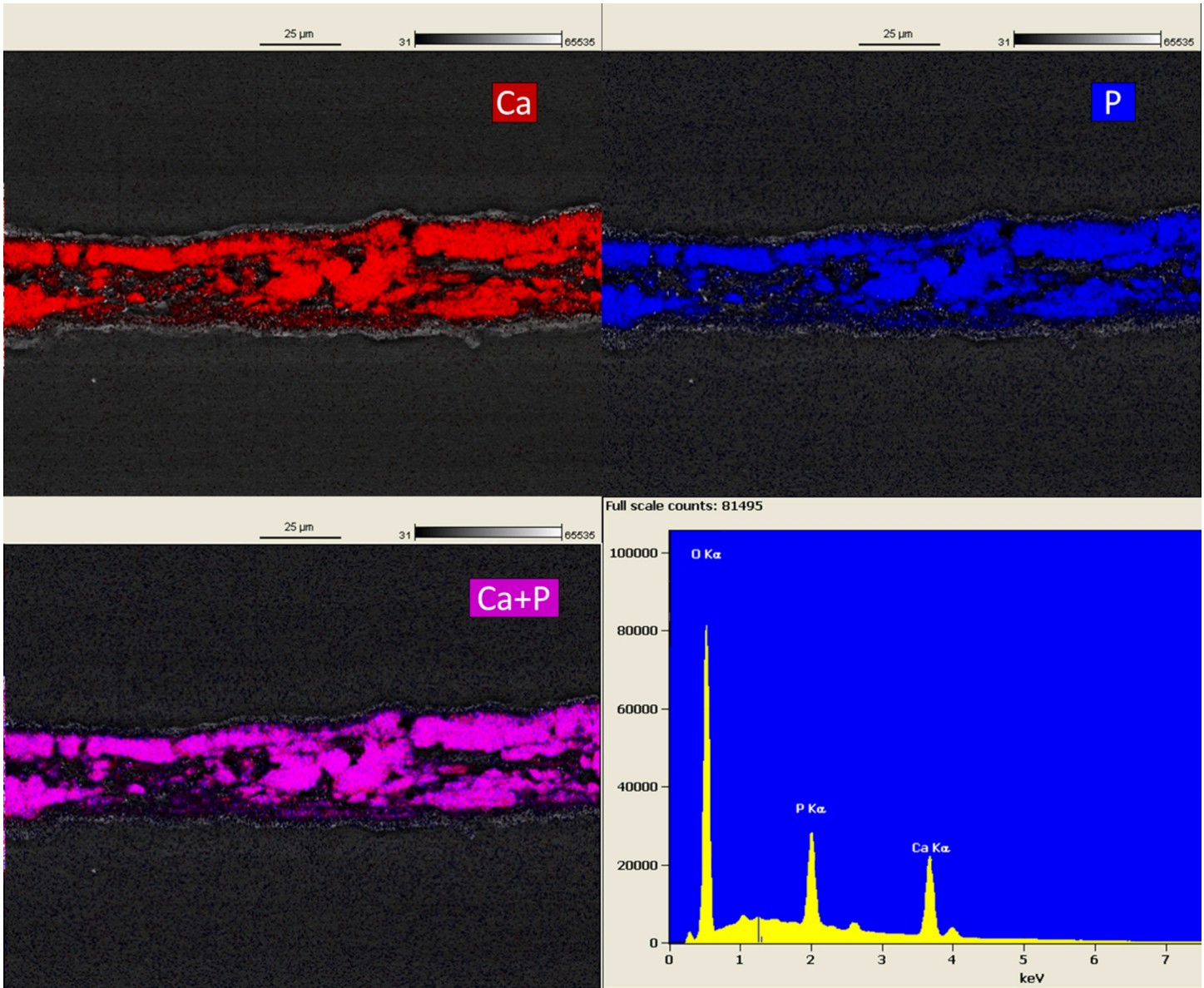

**Fig 14. SEM-EDS maps and spectrum from the same section of a no-PEMF control ring construct in Fig 12 with emphasis on Ca, P localization, and their overlay.** Ca/P ratio = 1.29. SEM settings: SEM settings: EHT = 15KeV, WD = 11.0 mm, H = 128.3 μm, W = 171.0 μm.

content and density, it is likely that the ring mineral/matrix ratio would more closely approach the control calvaria with continued culture. Parallel analysis of mineral phase distribution and hydroxyapatite content by XRD demonstrated that hydroxyapatite represented 50% of the no-PEMF control mineral and 80% of mineral from calvaria and femur. The hydroxyapatite content of ring mineral may be similar to newly formed mineral *in vivo*, which is predominantly calcium-deficient hydroxyapatite, and also carbonate-substituted hydroxyapatite [41, 42]. Since both forms can evolve under the influence of temperature (here during TGA analysis, 800˚C) to tricalcium phosphate rather than hydroxyapatite [34, 41], the hydroxyapatite content in culture actually may be higher than analyzed. However, in either circumstance, the mineral can be characterized as less mature than that from the 6-month-old control femur, consistent with relative crystallite sizes of 29.3 nm for the femur (S6 Fig) and 16.4 nm and 16.8

for no-PEMF and PEMF ring culture, respectively (S7 Fig). The presence of calcium-deficient hydroxyapatite was also suggested by the Ca/P ratios derived from SEM-EDS of ring cultures (Figs 13 and 14) that were within the range reported for young mouse calvaria [37], and by the even distribution of dense mineral accumulation domains by backscattered electron imaging. Neither of these parameters provides an unbiased quantitative assessment of mineral stoichiometry. In particular, the former is sensitive to the greater x-ray production volume of phosphorus x-rays than for calcium [43]. Further culture allowing ionic substitution to the mineral lattice or exposure to bone remodeling *in vivo* would be expected to further increase the maturity of such hydroxyapatite crystal structures. Thus, collectively, these data support the utility of the ring culture system to evaluate biomineralization as a controlled, progressive process in the investigation of the osteogenic effects of PEMF.

## PEMF effects

PEMF stimulates osteogenesis/mineralization in a variety of circumstances in the ring culture system. The most instructive circumstance was when PEMF was applied on Day 11 (Protocol B) (Fig 5), coincident with the change to Mineralization Medium, which provided for the first time an adequate phosphate source (3 mM beta-glycerol phosphate) to permit the deposition of mineral and ARS to measure it. Importantly, this was preceded by 10 days of culture in Differentiation Medium that drives osteoblast differentiation and the deposition of an extensive collagenous extracellular matrix, required as a template for mineral nucleation and accretion. Following a 3-day lag to Day 14, PEMF demonstrated a rapid increase in the mineral deposition, resulting in an approximately 4-fold higher ARS fluorescence than in the no-PEMF controls on Day 16. After Day 18, PEMF-treated cultures no longer demonstrated an enhanced rate of mineral deposition but accumulated ARS fluorescence at a rate similar to no-PEMF controls (Fig 5). Thus, the effect of PEMF treatment can be characterized in this protocol (B) as a lag, followed by enhanced mineralization, followed by cessation of stimulation of mineralization within 7 days. This character was specific to PEMF treatment since the cessation of stimulation has no parallel in the rate of mineralization in no-PEMF controls.

Insight into the processes involved in PEMF stimulation of mineralization can be gained by comparing the results of the other PEMF treatment protocols. In Protocol A, with PEMF treatment between D1 and D10, mineralization was rapid and to the same extent during the measurement window (D11-D18) as in Protocol B, even though no PEMF was administered during that window in Protocol A (Fig 5). In Protocol A, mineralization began earlier at D14 with a decreased lag period. Thus, cultures were able to respond to PEMF in the first 10 days of cultures in a way that was stable and not lost during the absence of PEMF after Day 10. These changes also advanced the PEMF-responsive process so that mineralization could begin sooner after switching to Mineralization Medium. Daily PEMF (Protocol D) gave the same early response and magnitude of stimulation as Protocol A because both started PEMF at D1 (Fig 5); however, PEMF treatment continued after D10 without any additional stimulation of mineralization. Thus, cessation of PEMF responsiveness after D18 is due to a loss of capacity rather than the absence of a PEMF signal. Finally, different than the stability of the effects of PEMF during culture (i.e., Protocol A), the culture maintains its capacity to respond to PEMF until late in culture, as demonstrated by the results of Protocol C. Here PEMF treatment was delayed until Day 28 but resulted in a similar increase in mineralization as seen with PEMF treatment earlier in culture (Fig 4C and 4B). Thus, the capacity and the limits of the capacity for PEMF to stimulate mineralization are maintained during 3 weeks of osteocyte differentiation and mineralization in the control environment, when it might be expected that these ongoing processes would consume resources necessary for the complete PEMF response.

The micro-CT data support these conclusions on stimulation of the amount of mineral deposited in response to PEMF and demonstrate that PEMF caused an increase in the rings' mineral density. However, PEMF did not negatively change the character or distribution of mineral as judged by mineral/matrix ratio or hydroxyapatite content, and thus, the mineralized matrix should be functional in an *in vivo* environment.

These characteristics of the response to PEMF in ring culture are consistent with a PEMF-dependent enhancement of osteocyte differentiation. There is a time-dependent association between colocalized mineral deposition and expression of the osteocyte marker, *Dmp1*-driven fluorescent reporter [13, 30]. Differentiation into an osteocyte and its variety of functions is multi-step [44] as indicated by the sequential overlapping expression of *Dmp1*, E11, PHEX, MEPE, sclerostin, and FGF21 [13], and PEMF intervention at any of these or other steps could likely speed the overall deposition of mineral. Recently, genetic deletion of EphrinB2 in early-stage osteoblasts using *Osx1Cre* led to delayed initiation of mineralization but normal mineral accumulation. However, deletion in late osteoblast/early osteocytes with *Dmp1Cre* led to normal initiation but increased mineral accumulation and maturation [15, 45]. Thus, small changes in the timing and environment of gene expression can cause independent regulation of initiation and accumulation of mineral [15]. These experiments may serve as a conceptual example of how PEMF stimulates mineralization.

A detailed analysis of mineral deposited in monolayer culture by the mouse preosteoblastic clonal cell line, MC3T3-E1, demonstrated mineral characteristics similar to those in one-month-old mouse calvaria using techniques like the ones we have used: XRD, SEM-EDS, SEM, as well as FTIR and TEM [46]. Our results, including those from TGA and micro-CT, agree that bone-like mineral can be produced in culture. However, there are several limitations to using the MC3T3 system for assessing treatments for osteogenesis. Principally, the mineral deposition was sparse, indicating that many cells that were phenotypically capable of mineral deposition were not engaged in that process. Thus, a low response signal was generated in the presence of a high non-responsive background. This emphasizes the importance of cell-matrix and cell-cell interactions. In ring culture, these are rapidly enhanced by the matrix contraction that occurred early in culture, placing cells in close proximity to each other and the transient fibrin matrix and resulting in a bone-like pattern of dense mineral deposition when frozen sections were viewed by fluorescence microscopy (S9 Fig). There was also no exposure to the differentiation-disruptive signaling induced by plastic attachment that predominates in monolayer culture, at least until multilayers are formed. In addition, the use of a clonal line limits the diversity of lineage commitment and progression that will be screened for the response to treatment. Primary cultures and mesenchymal stem cells provide more relevant targets for osteogenesis in preclinical testing.

Although ring culture is limited by the necessity of production of rings, casting trays, and ring supports, it enables a realistic osteogenic process in three dimensions that can be further stimulated by PEMF treatment both early and late in culture. Ring culture also provides several advantages for additional investigation of the effects of PEMF on osteogenesis. A narrow 5-day window of PEMF treatment, from Day 11 to Day 16, exhibits a highly significant 4-fold increase in the mineral deposition. This is a large enough difference to provide interpretable dissection of PEMF signaling mechanisms and pathways. Optimization of PEMF parameters and dose also will be facilitated in this short window. In addition, PEMF treatment in the first 10 days provides a more mechanism-selective environment since mineralization will not be present at the same time. This allows the PEMF treatment period to be shortened to find its minimum before allowing mineralization after day 11 as the outcome measure. Using a short PEMF exposure time (possibly only one day) while monitoring signaling pathways in the presence of agonists and antagonists will provide greater precision and variety in screening for mechanism and PEMF parameters.

The current results with ring culture also inform the application of PEMF in the clinical situation. Ring culture provides a defined number and character of precursor cells simultaneously exposed to signals for osteoblast differentiation at initiation and a culture duration that allows progression through osteocyte differentiation and mineral maturation. Although these processes overlap, the application of PEMF in the intervals defined by Protocols A-D, allows the determination of the effectiveness of PEMF to influence these processes. That PEMF produced its most rapid and largest stimulation of mineralization assessed by both ARS fluorescence and micro-CT (Figs 5 and 6) during periods of osteoblast and osteocyte differentiation (Protocols A, B, and D) suggests that it would have its greatest positive impact clinically if applied soon after fracture when these processes dominate. Such early mineralization may lead to matrix stabilization and help prevent non-unions.

In contrast, that PEMF stimulation was possible later in culture in Protocol C suggests that cells in the original potentially responsive population retain that responsiveness during active osteogenesis. Thus, resting precursors and precursors recruited later in the *in vivo* repair process will likely respond to PEMF. This would be especially important when treating non-unions where considerable time has elapsed following fracture before PEMF treatment, and the exact cause of impaired repair may not be known. Indeed, the PEMF effect *in vivo* may be greater than that described in this study because of the opportunity for continued recruitment of responsive precursors from the circulation and proximal tissue, processes inherently absent during ring culture. This study is thus a first step in exploring the potential effects of timing of PEMF treatment during repair and may influence clinical PEMF treatment strategies.

PEMF optimization may lead to more effective treatment of a variety of musculoskeletal ailments, in particular those involving bone metabolism [1]. The osteogenic ring culture system is expected to facilitate this endeavor as well as contribute to preclinical screens of biologic and pharmacologic agents targeted at bone deposition, turnover, and resorption. Additionally, ring culture may enhance investigation in other experimental systems that benefit from the 3D culture of single or multiple cell types but are limited by the geometry of spheroid or organoid culture such as metastasis, vasculogenesis, and skin equivalents [47].

## Supporting information

**S1 Fig.** Mineralization measured by ARS (a) and by micro-CT (b) at Day 63. For ARS (a) P = Tukey HSD, groups not significantly different. For micro-CT (b), n = 4 per group, except Day 1-Day 11 where n = 3. Significant stimulation was detected for PEMF from Day 28-Day 63, and exposure to Daily PEMF.
(ZIP)

**S2 Fig. Micro-CT-derived density profiles at termination, Day 63.** Total HA *vs*. HA/ml was plotted and profiles between no-PEMF controls and PEMF treatment between Day 1- Day 10 compared (a); PEMF between Day 11-Day 27 (b); PEMF between Day 28-Day 63 (c); and Daily PEMF (d). In each case n = 3, the line is the mean and its thickness represents the standard error. Control values were also subtracted from PEMF values at each density to yield a difference plot (orange) to visualize the effect of PEMF on mineral density.
(ZIP)

**S3 Fig. Comparing micro-CT mineral density profiles from calvaria, no-PEMF, and Daily PEMF ring constructs on Day 63.** The mineral mass at each density was converted to a percentage of the total mass of the imaged construct/calvaria to allow easy comparison of the samples. For ring constructs n = 3, and the plotted line is the mean; for calvaria, n = 1.
(TIF)

**S4 Fig.** Computed cross-sections of micro-CT images on Day 63 of no-PEMF control constructs (a), Daily PEMF constructs (b), and 6-month-old calvaria (c). Sections were 10 microns thick.
(TIF)

**S5 Fig. Automated temperature and weight measurement program for thermogravimetric analysis to obtain mineral/matrix ratios.**
(TIF)

**S6 Fig. XRD reflections and HA content pie-chart of a TGA processed mouse femur.** Mean crystallite size is in the top left corner.
(TIF)

**S7 Fig. XRD reflections and HA content pie charts of TGA processed ring constructs cultured without PEMF (top panel) or treated with PEMF on Day 28-Day 63 (bottom panel).** Mean crystallite size is in the top left corner.
(TIF)

**S8 Fig. SEM image of the top surface of a Daily PEMF construct after processing for collagen visualization as in Fig 2A.** A fine mesh of collagen fibrils without large fibers contrasted with the deposited collagen architecture of the construct interior (Fig 2A). SEM settings: EHT = 1 KeV, WD = 2.4 mm, H = 17.42 μm, W = 23.23 μm, Mag = 4.92 K X. Scale Bar = 1 μm.
(TIF)

**S9 Fig. 3D osteoblast/osteocyte ring cultures from a previous study terminated after 34 days of culture with estradiol (+E) or without (Control) and visualized by fluorescence microscopy of frozen sections for mineral (red, ARS), alkaline phosphatase (green, ELF-97), and DNA (blue, Hoechst 33348).** Note the increased continuity of mineral deposition and increased expression of alkaline phosphatase activity as a marker of the osteoblast lineage in the presence of estradiol, present in both Differentiation and Mineralization Media.
(TIF)

**S10 Fig. Apparatus for exposure of ring cultures to PEMF.** Note the rectangular culture treatment areas marked by white borders on each shelf. The center two shelves were routinely used.
(TIF)

## Acknowledgments

The authors wish to thank the Molecular Imaging Center core in the Department of Radiology, Keck School of Medicine, University of Southern California and Mr. Tautis Skorka for assistance, consulting and performing high-resolution micro-CT scanning of the samples for this study using a Scanco μCT50 micro-CT system (http://www.scanco.ch/) at MIC (http://mic.usc.edu/). We thank Volume Graphics, GmbH (Heidelberg, Germany) for providing complimentary access to VGSTUDIO MAX 3.3.2 used in this work (https://www.volumegraphics.com). We thank and acknowledge the contributions from Dr. Saeed I. Khan at the UCLA Molecular Instrument Center for X-ray diffraction analysis of our samples. We also acknowledge access and consulting with the UCLA Architecture Workshop for the design and manufacture of the Teflon rings, supports, and casting trays.

## Author Contributions

**Conceptualization:** Paul D. Benya, Nianli Zhang, Erik I. Waldorff, James T. Ryaby, Fabrizio Billi.

**Data curation:** Aaron Kavanaugh.

**Formal analysis:** Aaron Kavanaugh, Tea Jashashvili.

**Funding acquisition:** Nianli Zhang, Fabrizio Billi.

**Investigation:** Paul D. Benya, Aaron Kavanaugh, Martin Zakarian, Philip Söderlind, Tea Jashashvili.

**Methodology:** Paul D. Benya, Aaron Kavanaugh, Philip Söderlind, Fabrizio Billi.

**Project administration:** Paul D. Benya, Fabrizio Billi.

**Resources:** Paul D. Benya, Fabrizio Billi.

**Software:** Aaron Kavanaugh.

**Supervision:** Paul D. Benya, Fabrizio Billi.

**Validation:** Paul D. Benya, Aaron Kavanaugh, Fabrizio Billi.

**Visualization:** Paul D. Benya, Aaron Kavanaugh, Martin Zakarian, Fabrizio Billi.

**Writing – original draft:** Paul D. Benya, Fabrizio Billi.

**Writing – review & editing:** Paul D. Benya, Aaron Kavanaugh, Tea Jashashvili, Nianli Zhang, Erik I. Waldorff, James T. Ryaby, Fabrizio Billi.

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
