## [Decision Letter · Decision Letter 0]

16 Oct 2020

PONE-D-20-27626

Pulsed electromagnetic frequency (PEMF) transiently stimulates the rate of mineralization in a 3-dimensional ring culture model of osteogenesis

PLOS ONE

Dear Dr. BILLI,

Thank you for submitting your manuscript to PLOS ONE. After careful consideration, we feel that it has merit but does not fully meet PLOS ONE’s publication criteria as it currently stands. Therefore, we invite you to submit a revised version of the manuscript that addresses the points raised during the review process.

Your manuscript has been evaluated by two experts in the field and by a senior member of our Editorial Board.  I am pleased to inform you that, while there is recognition and appreciation for the results that you present in your manuscript, there is a requirement for revisions before the paper can be accepted for publication in PLOS ONE. 

Reviewer number one recommends several modifications in the presentation and interpretation of your findings.  To address the concerns of reviewer number two it is necessary to elaborate on the relevance of your results to advancing understanding of bone biology and pathology and to justify publication of your paper in PLOS ONE rather than in a specialized biomaterials journal.

We thank you for submitting your paper to PLOS ONE and look forward to receiving a revised manuscript in which recommendations that were raised during the initial review have been addressed.

We look forward to receiving your revised manuscript.

Kind regards,

Gary Stein

Academic Editor

PLOS ONE

Journal Requirements:

2. At this time, we request that you  please report additional details in your Methods section regarding animal care, as per our editorial guidelines: 1) Please provide details of animal welfare (e.g., shelter, food, water, environmental enrichment), 2) please report the number of animals used in this study, the genetic strain and source of the mice. Thank you for your attention to these requests.

3. To comply with PLOS ONE submission guidelines, in your Methods section, please provide additional information regarding your statistical analyses.  Specifically, please describe any statistical tests used in the Methods. For more information on PLOS ONE's expectations for statistical reporting, please see https://journals.plos.org/plosone/s/submission-guidelines.#loc-statistical-reporting

'NZ, EIW, and JTR are employees of Orthofix, Inc. PDB, AK, and FB received salary support from Orthofix, Inc. TJ and PS received fee-for-service support from Orthofix, Inc. Publication fees were paid by Orthofix. PDB and FB received support to attend meetings and present data. FB received salary and general support from the Orthopaedic Institute for Children. Participation in this research by Orthofix and the Orthopaedic Institute for Children is described in the Financial Disclosure Statement.

There are no patents, products in development, or marketed products to declare regarding the data in this publication related to the authors or either funding institution.

Orthofix, Inc. markets medical devices that apply PEMF in clinical practice.'

Reviewers' comments:

Reviewer's Responses to Questions

**Comments to the Author**

1. Is the manuscript technically sound, and do the data support the conclusions?

Reviewer #1: Yes

Reviewer #2: Yes

2. Has the statistical analysis been performed appropriately and rigorously? 

Reviewer #1: I Don't Know

Reviewer #2: Yes

3. Have the authors made all data underlying the findings in their manuscript fully available?

Reviewer #1: Yes

Reviewer #2: Yes

4. Is the manuscript presented in an intelligible fashion and written in standard English?

Reviewer #1: Yes

Reviewer #2: Yes

5. Review Comments to the Author

Reviewer #1: The authors examined the effects of PEMF on osteogenesis in a 3-D ring culture model with different PEMF regimens over 62 days in culture in osteogenic enriched media. Day 2 mouse calvarial pre-osteoblasts were cast around Teflon rings by polymerization of fibrinogen (no tissue culture plastic contact) and high frequency range of PEMF regimen (42.85kHz, 67ms burst) was applied to cells over a 4 hour period. This regimen either spanned the duration of time points, or was used from D1-D10, D11-D27, or D28-D63.

Outcomes measures included osteogenesis (Alizarin Red S), which was kinetically measured with micro-CT correlates. Scanning electron microscopy and elemental analyses (EDS) were also performed.

The authors' experiments found that D1-D10 and D11-D27 PEMF treatment increased mineralization 4, while D28-42 stimulation only significantly increased micro-CT values. Further, PEMF shifted micro-CT densities to higher profiles of mineral matrix via thermogravimetric analysis

These results suggest that osteogenic induction can support late-stage PEMF induction to be effective.

This is a thorough, interesting study that provides value to the literature. There are however some experimental details which should be clarified and limitations which need to be discussed in this manuscipt.

1. The authors do not adequately describe the neonatal mice they used. What genetic background were these mice? Mineralization can be effected by genetic background.

2. The statistics are not adequately described. The Methods section does not provide any details and references the Figures. One Figure sites the Tukey test with no rationale. P-values are provided, but the Methods section must thoroughly outline specific statistics and rationale for their use.

Regrading the limitations of this manuscript:

1. This was an in-vitro study which induced osteogenesis in order to assess mineralization. It is known that in the clinical setting, the effectiveness external treatment regimens are highly dependent on patient compliance. Further, effects on bone healing are limited in vivo after the early stages of bone repair. An in vitro condition with factors which continually promote osteogenesis is not translatable to a patient when inflammatory factors subside.

2. Since the authors goal was to examine the effectiveness of late-stage PEMF on mineralization in with sustained osteogenesis, a molecular/cellular outcome would have been informative relating to any potential mechanism of late-stage PEMF on this matrix.

Reviewer #2: The manuscript by Benya, et al., entitled “Pulsed electromagnetic frequency (PEMF) transiently stimulates the rate of mineralization in a 3-dimensional ring culture model of osteogenesis” presents the results of experiments designed to ascertain the mechanism/outcome of the delivery of a pulsed electromagnetic field on the process of steogensis, at least as we understand the process in vitro. The experiments utilize a conventional calvaria-derived osteoblastogenic assay, that has been modified (in an innovative way) via the incorporation of a three-dimensional ring culture system. The results suggest that there is a profound effect of PEMF on the generation of collagen and mineral matrix.

Concerns:

• It is difficult to determine with precision the “dose” of PEMF delivered? Clinically, the use of PEMF is controlled both temporally and geographically. The dosing regimen “4h/day for the entire culture (Daily), or just during Day1-Day10, Day11-Day 27, or Day28-Day63 and cultured without PEMF for the preceding or remaining days, and compared to no-PEMF controls” would benefit from some rationale and discussion that aligns the investigators thinking with clinical use of PEMF.

• Is the frequency of 40.85 kHz consistent with clinical use?

• The data suggest some aspects of mineral material properties are consistent with de novo bone or mineral, yet no evidence of cellular function or indeed whether the cells are osteoblastic in nature vs. fibroblastic?

• Is there any histological evidence to directly demonstrate cellular function and mineralization in vitro?

• What are PEMF “cellular receptors”? Explain and provide rationale

• It remains unclear how the extent of mineralization actually correlates with de novo bone formation in vivo, hence the conclusions suggesting the association of mineral content and alizarin staining with any kind of bone formation should be tempered

• It is critical to link the in vitro studies with something of physiologic relevance, the idea of which is completely lacking in the manuscript.

6. PLOS authors have the option to publish the peer review history of their article (what does this mean?). If published, this will include your full peer review and any attached files.

Reviewer #1: No

Reviewer #2: No

---

## [Author Response · Author response to Decision Letter 0]

28 Nov 2020

Response to Reviewers

Journal Requirements:

1. Please ensure that your manuscript meets PLOS ONE's style requirements, including those for file naming. Response: The manuscript meets the style requirements. 

2. At this time, we request that you please report additional details in your Methods section regarding animal care, as per our editorial guidelines: 1) Please provide details of animal welfare (e.g., shelter, food, water, environmental enrichment), 2) please report the number of animals used in this study, the genetic strain and source of the mice. Thank you for your attention to these requests. Response: See below for response to Reviewer comment. 

3. To comply with PLOS ONE submission guidelines, in your Methods section, please provide additional information regarding your statistical analyses. Specifically, please describe any statistical tests used in the Methods. Response: See below for response to Reviewer comment. 

 Response: Below is our revised Competing Interest section which complies with the PLOS One data sharing requirement and declaration of competing interests. It is also present in our Cover Letter.

'Competing Interests: The authors have read the journal’s policy and the authors of this manuscript have the following competing interests: NZ, EIW, and JTR are employees of Orthofix, Inc. PDB, AK, and FB received salary support from Orthofix, Inc. TJ and PS received fee-for-service support from Orthofix, Inc. Publication fees were paid by Orthofix. PDB and FB received support to attend meetings and present data. FB received salary and general support from the Orthopaedic Institute for Children. Participation in this research by Orthofix and the Orthopaedic Institute for Children is described in the Financial Disclosure Statement.

There are no patents, products in development, or marketed products to declare regarding the data in this publication related to the authors or either funding institution.

Orthofix, Inc. markets medical devices that apply PEMF in clinical practice. This does not alter our adherence to PLOS ONE policies on sharing data and materials.’ 

5. We note that you have stated that you will provide repository information for your data at acceptance. Should your manuscript be accepted for publication, we will hold it until you provide the relevant accession numbers or DOIs necessary to access your data. 

Response: No changes in our Data Availability statement were needed or made.

Reviewers' comments:

Reviewer's Responses to Questions

Comments to the Author

1. Is the manuscript technically sound, and do the data support the conclusions?

Reviewer #1: Yes

Reviewer #2: Yes

2. Has the statistical analysis been performed appropriately and rigorously? 

Reviewer #1: I Don't Know

Reviewer #2: Yes

 3. Have the authors made all data underlying the findings in their manuscript fully available?

Reviewer #1: Yes

Reviewer #2: Yes

 4. Is the manuscript presented in an intelligible fashion and written in standard English?

 Reviewer #1: Yes

Reviewer #2: Yes

 5. Review Comments to the Author

Reviewer #1: The authors examined the effects of PEMF on osteogenesis in a 3-D ring culture model with different PEMF regimens over 62 days in culture in osteogenic enriched media. Day 2 mouse calvarial pre-osteoblasts were cast around Teflon rings by polymerization of fibrinogen (no tissue culture plastic contact) and high frequency range of PEMF regimen (42.85kHz, 67ms burst) was applied to cells over a 4 hour period. This regimen either spanned the duration of time points, or was used from D1-D10, D11-D27, or D28-D63.

Outcomes measures included osteogenesis (Alizarin Red S), which was kinetically measured with micro-CT correlates. Scanning electron microscopy and elemental analyses (EDS) were also performed.

The authors' experiments found that D1-D10 and D11-D27 PEMF treatment increased mineralization 4, while D28-42 stimulation only significantly increased micro-CT values. Further, PEMF shifted micro-CT densities to higher profiles of mineral matrix via thermogravimetric analysis

These results suggest that osteogenic induction can support late-stage PEMF induction to be effective.

This is a thorough, interesting study that provides value to the literature. There are however some experimental details which should be clarified and limitations which need to be discussed in this manuscipt.

1. The authors do not adequately describe the neonatal mice they used. What genetic background were these mice? Mineralization can be effected by genetic background. 

Response: This information has been added by inclusion of the following text in Methods: “Timed-pregnant 022-CD-1 mice were obtained from Charles River Laboratories (Wilmington, MA 01887 US), delivered on E13 and individually housed in micro-isolator cages enriched with nestlet. Preosteoblasts were isolated from 160 postnatal day 2 mouse calvaria following dissection of the parietal bones without sutures.“

2. The statistics are not adequately described. The Methods section does not provide any details and references the Figures. One Figure sites the Tukey test with no rationale. P-values are provided, but the Methods section must thoroughly outline specific statistics and rationale for their use. 

Response: The following revised Statistics section has been included in the Methods section.

Statistical comparisons were performed in Microsoft Excel (Redmond, Washington) augmented by the Real Statistics Resource Pack software for Excel (Release 6.8), Copyright (2013 – 2020) Charles Zaiontz. www.real-statistics.com. Data for micro-CT analysis was extracted from DICOM images using a custom MATLAB program and subsequently transferred to Excel. α = 0.05 was chosen as the threshold of statistical significance for all analysis. Statistical comparisons were made pairwise between the control group (No PEMF) and each of the other test groups (D1-D10, D11-D27, and D28-Term) for all comparisons to assess the effectiveness of PEMF at each time interval versus an absence of PEMF. Tukey HSD tests were chosen for all comparisons because these control for family-wise type 1 error across multiple pairwise comparisons. Error bars, for those figures which possess them (Figures: 6a, b; 9, 10, S1a, b) indicate standard error. Error bands (Figures: 7a, b, c, d and S2a, b, c, d) likewise indicate standard error. In Figures 7a, b, c, d and S2a, b, c, d, which display micro-CT total µg HA vs HA mg/mL, data is smoothed with a 5x1 moving average filter. 

Regarding the limitations of this manuscript:

1. This was an in-vitro study which induced osteogenesis in order to assess mineralization. It is known that in the clinical setting, the effectiveness external treatment regimens are highly dependent on patient compliance. Further, effects on bone healing are limited in vivo after the early stages of bone repair. An in vitro condition with factors which continually promote osteogenesis is not translatable to a patient when inflammatory factors subside.

Response: Patient compliance in the clinical setting will always be an issue regardless of whether the treatment is external or therapeutic. However, PEMF has the potential to influence in vivo osteogenesis with minimal side effects and warrants investigation to maximize the opportunity of this treatment modality. In this regard, optimization of PEMF characteristics may lead to treatment regimens allowing higher rates of compliance and/or clinical efficacy. From the original manuscript regarding the multiple PEMF treatment periods “This permitted determination of whether PEMF selectively enhanced any stage of osteogenesis.” Thus, both early and late stages of osteogenesis were evaluated. Indeed, the early PEMF effects (4-fold, D12-D18, Procedure B) would likely be applicable and effective in the early clinical period of osteogenesis before subsidence of the inflammatory factors referred to by Reviewer #1. This is particularly important because the PEMF stimulation was observed in vitro without the need of these inflammatory factors and vascular and other cell types influencing the hormonal milieu. Thus, we are evaluating the direct effect of PEMF on the central (not the only) cellular mediators of osteogenesis, namely preosteoblasts, osteoblasts, and osteocytes. The effectiveness of PEMF without inflammatory factors at least suggests that it may extend the period of repair beyond the usual duration marked by the decline in inflammatory factors. The intent and result of the data presented was to describe a robust culture model for investigation of PEMF characteristics and mechanisms, not an evaluation of mechanisms themselves. The data permitted proposal in the Discussion of experimental designs providing documented 4-fold enhancement by PEMF over only 5 days suitable for detailed optimization of in vitro PEMF treatment that will justify evaluation of new parameters in vivo, an established preclinical approach. Whether the effects of PEMF observed in this in vitro study will benefit a patient in vivo is an entirely separate question from whether PEMF causes beneficial changes in fundamental processes of osteogenesis in vitro, the intent of this work. In vitro optimization should precede taking this treatment modality to animal models to substantiate or refute the reviewer’s supposition that PEMF will not be beneficial in vivo. Indeed, clinical evaluation of some forms of PEMF have demonstrated utility in facilitating bone repair, in particular non-unions where the reviewer would have expected no benefit because of the long time elapsed after fracture, as referenced in the Introduction. Developing clinically beneficial treatments is a multi-step process, not accomplished all at once. We believe this work substantially contributes novel observations using a sophisticated approach that establishes a basis for further evaluation and optimization of PEMF before progressing to animal models or clinical trials.

2. Since the authors goal was to examine the effectiveness of late-stage PEMF on mineralization in with sustained osteogenesis, a molecular/cellular outcome would have been informative relating to any potential mechanism of late-stage PEMF on this matrix.

Response: In actuality, the authors’ goal was to evaluate the effectiveness of PEMF on any of the investigated stages of osteogenesis as discussed above and presented in the original manuscript. We have used the integrative outcome measure of mineralization and mineral character to monitor osteogenesis. Indeed, mineralization is “a molecular/cellular outcome measure” since it depends on prior osteoblast and osteocyte differentiation as demonstrated by Dallas (see ref below) using a highly similar calvarial preosteoblast monolayer culture model of osteogenesis which was cited multiple times in the original manuscript. In this work preosteoblasts were derived from genetic knock-in newborn mice that express GFP under the control of the DMP-1 promoter. Thus, the expression of GFP is a surrogate of DMP-1 expression, an acknowledged molecular marker of osteocytes. When GFP and ARS mineral fluorescence were simultaneously measured during osteogenesis in monolayerculture by dual channel time-lapse microscopy these signals were parallel in induction and intensity and colocalized. This demonstrates that only cells expressing an osteocyte phenotype were responsible for mineral deposition, not fibroblasts or undifferentiated cells. Consequently, mineralization measures these cellular differentiation processes as well as reflects the functional consequence of mineralized matrix deposition essential for bone. Again, the intent of the manuscript was the outcome effect of PEMF, not discerning mechanisms, which will be the focus of future directed experimentation. 

The following modified paragraph has been included in the Discussion.

 “This conclusion is supported by the work of Dallas (Dallas, SL) using a highly similar calvarial preosteoblast, but monolayer, culture model of osteogenesis with preosteoblasts that express GFP under the control of the DMP-1 promoter. Thus, expression of GFP in this system was a surrogate of DMP-1 expression, an acknowledged molecular marker of osteocytes. When GFP and ARS mineral fluorescence were simultaneously measured during osteogenesis in culture by dual channel time-lapse fluorescent microscopy these signals were parallel in induction and intensity and colocalized. This demonstrates that only cells expressing an osteocyte phenotype were responsible for mineral deposition, not fibroblasts or undifferentiated cells. Similarly, in vitro osteogenesis of the clonal cell line, IDG-SW3 produced parallel ARS-detected mineral deposition and DMP-1 expression with colocalization. This pattern closely resembled expression of other acknowledged early osteocytic markers E11, MEPE, PHEX and the early osteoblast marker, alkaline phosphatase (Woo, S.M). Consequently, mineralization measures these cellular differentiation processes as well as reflects the functional consequence of mineralized matrix deposition essential for bone.”

Dallas SL, Veno PA. Live imaging of bone cell and organ cultures. In: Helfrich MH, Ralston SH, editors. Bone Research Protocols. Totowa, NJ: Humana Press; 2012. pp. 425–457. doi:10.1007/978-1-61779-415-5_26

Woo SM, Rosser J, Dusevich V, Kalajzic I, Bonewald LF. Cell line IDG-SW3 replicates osteoblast-to-late-osteocyte differentiation in vitro and accelerates bone formation in vivo. J Bone Miner Res. 2011;26: 2634–2646. doi:10.1002/jbmr.465

Reviewer #2: The manuscript by Benya, et al., entitled “Pulsed electromagnetic frequency (PEMF) transiently stimulates the rate of mineralization in a 3-dimensional ring culture model of osteogenesis” presents the results of experiments designed to ascertain the mechanism/outcome of the delivery of a pulsed electromagnetic field on the process of steogensis, at least as we understand the process in vitro. The experiments utilize a conventional calvaria-derived osteoblastogenic assay, that has been modified (in an innovative way) via the incorporation of a three-dimensional ring culture system. The results suggest that there is a profound effect of PEMF on the generation of collagen and mineral matrix.

Concerns:

• It is difficult to determine with precision the “dose” of PEMF delivered? Clinically, the use of PEMF is controlled both temporally and geographically. The dosing regimen “4h/day for the entire culture (Daily), or just during Day1-Day10, Day11-Day 27, or Day28-Day63 and cultured without PEMF for the preceding or remaining days, and compared to no-PEMF controls” would benefit from some rationale and discussion that aligns the investigators thinking with clinical use of PEMF. 

Response: 

The following text has been included in Methods.

“COMSOL Multiphysics software was used to simulate the distribution of EMF in the supporting rack. EMF distribution was also verified with a Hall Effect transverse Gaussmeter probe (FW Bell 5186 Gauss Meter equipped with a SAD18-1904 axial probe, Berg Engineering, Rolling Meadows, IL) while the temporal pattern of the electromagnetic signal was evaluated by a digital oscilloscope (TBS1000C, Tektronix, Beaverton, OR). The magnetic field intensity was measured using the Gaussmeter probe placed on the rack at three different positions, center, 8 cm from the center, and 1 cm from the edge. The distribution of magnetic field intensity in the rack allowed for definition of a safe zone around the center where the EMF was deemed constant. The cell culture plates were placed in this area marked by white rectangular borders (S10 Fig). Both the simulation model and the probe verification demonstrated that the cells were exposed to a magnetic field varying from 1.14 to 1.24 mT depending on the proximity to the rack center, with 1.19 mT as the average.”

The following modified paragraph has been included in the Discussion.

“The current results with ring culture also inform the application of PEMF in the clinical situation. Ring culture provides a defined number and character of precursor cells simultaneously exposed to signals for osteoblast differentiation at initiation and a culture duration that allows progression through osteocyte differentiation and mineral maturation. Although these processes are overlapping, application of PEMF in the intervals defined by Protocols A-D, allows determination of effectiveness of PEMF to influence these processes. That PEMF produced its most rapid and largest stimulation of mineralization assessed by both ARS fluorescence and micro-CT (Fig 5 and 6) during periods of osteoblast and osteocyte differentiation (Protocols A, B, and D) suggests that it would have its greatest positive impact clinically if applied soon after fracture when these processes dominate. Such early mineralization may lead to matrix stabilization and help prevent non-unions. In contrast, that PEMF stimulation was possible later in culture in Protocol C, suggests that cells in the original potentially responsive population, retain that responsiveness during active osteogenesis. Thus, resting precursors and precursors recruited later in the in vivo repair process will likely respond to PEMF. This would be especially important when treating non-unions where considerable time has elapsed following fracture before PEMF treatment and the exact cause of impaired repair may not be known. Indeed, the PEMF effect in vivo may be greater than that described in this study because of the opportunity for continued recruitment of responsive precursors from the circulation and proximal tissue, processes inherently absent during ring culture. This study is thus a first step in exploring the potential effects of timing of PEMF treatment during repair and may influence clinical PEMF treatment strategies.”

• Is the frequency of 40.85 kHz consistent with clinical use? 

Response: The following modified text has been added to Methods section of the manuscript. “The PEMF signal (HF Physio, 40.85 kHz frequency with 67 ms burst period) used in the current study is derived from an already FDA approved PEMF signal indicated for long-bone non-union fractures (PhysioStim™ from Orthofix Medical Inc.) [32]. The number of pulses, burst frequency and signal amplitudes are also similar. Hence the pulse frequency and the signal characteristics are well within the range of what is being used in the clinical setting. This PEMF signal is used here to determine its potential benefit in an in vitro osteogenic environment and increase the diversity of PEMF parameters available for screening and optimization.” 

• The data suggest some aspects of mineral material properties are consistent with de novo bone or mineral, yet no evidence of cellular function or indeed whether the cells are osteoblastic in nature vs. fibroblastic?

• Is there any histological evidence to directly demonstrate cellular function and mineralization in vitro?

Response: We refer to the response above to Reviewer 1 part 2. “Indeed, mineralization is “a molecular/cellular outcome measure” since it depends on prior osteoblast and osteocyte differentiation as demonstrated by Dallas using a highly similar calvarial preosteoblast monolayer culture model of osteogenesis which was cited multiple times in the original manuscript. In this work preosteoblasts were derived from genetic knock-in newborn mice that express GFP under the control of the DMP-1 promoter. Thus, the expression of GFP is a surrogate of DMP-1 expression, an acknowledged molecular marker of osteocytes. When GFP and ARS mineral fluorescence were simultaneously measured during osteogenesis in culture by dual channel time-lapse microscopy these signals were parallel in induction and intensity and colocalized. This demonstrates that only cells expressing an osteocyte phenotype were responsible for mineral deposition, not fibroblasts or undifferentiated cells. Consequently, mineralization measures these cellular differentiation processes as well as reflects the functional consequence of mineralized matrix deposition essential for bone.” 

We have appended the following text to the above in the manuscript.

“Similarly, the time-course of ARS-detected mineral deposition matches that of DMP-1 and other acknowledged early osteocytic markers E11, MEPE, PHEX and the early osteoblast marker, alkaline phosphatase, during in vitro osteogenesis of the clonal cell line, IDG-SW3. This phenotypic transition to osteoblasts and mineralizing osteocytes is complex (Woo, S.M.) and not shared by fibroblasts or any other cell types.”

Yes, histological evidence to directly demonstrate cellular function and mineralization in vitro was provided in the original manuscript in Supplemental Figure 9. This figure demonstrates the selective cellular distribution of alkaline phosphatase, the early osteoblast differentiation marker, using its enzymatic activity with the ELF-97 fluorescent substrate (green) in frozen sections of rings labeled with intravital ARS staining for mineral deposition (red). Importantly, all cellular locations were identified by Hoechst 33348 nuclear staining (blue). Distinct alkaline phosphatase activity was present on the construct surface as well as the interior, demonstrating a robust commitment to osteoblast differentiation in this culture model. In contrast, extensive mineralization was restricted to the construct interior, a result entirely consistent with the mineral distribution demonstrated by secondary electron and backscatter SEM images in Figures 11, 12. Thus, this culture model generates constructs that resemble tissue organization with osteoblasts and osteoid at the surface and subjacent osteocytes embedded within a highly mineralized matrix. 

We have added such explanatory information at multiple places in the text and have moved the introduction of the figure to follow Figure 1 to add emphasis to this information.

• What are PEMF “cellular receptors”? Explain and provide rationale. 

Response: The term “PEMF cellular receptors” was used as a conceptual term. We agree that this may be confusing. We have replaced this with the term “direct molecular targets of PEMF. 

• It remains unclear how the extent of mineralization actually correlates with de novo bone formation in vivo, hence the conclusions suggesting the association of mineral content and alizarin staining with any kind of bone formation should be tempered

Response: De novo bone formation in vivo is routinely monitored during skeletogenesis and during the evaluation of genetically altered mice by embryonic whole mounts stained with alizarin red to mark bone. Similarly, gain- and loss-of-function mutants, central to our understanding of regulatory mechanisms in bone, are interpreted based on micro-CT imaging of bone content, bone volume, and bone mineral density. These approaches are based on the rate, extent, and morphology of deposited mineral, which are in turn dependent on the recruitment, activation, and progression of osteogenic precursors to osteoblasts and osteocytes. Indeed, deposition of an extensive collagen-rich mineralized bone matrix is a key functional endpoint in in vivo bone formation. We have used both alizarin red and micro-CT to measure mineral deposition and density in culture and mark the progression of precursor differentiation. In addition, we have demonstrated that the mineral deposited approaches the crystalline structure (hydroxyapatite) of in vivo bone with x-ray diffraction (62% of control bone) and the expected collagen content with TGA measurement of mineral:matrix ratio (80% of control bone) consistent with the activity of the osteogenic marker, alkaline phosphatase. Certainly, these characteristics of osteogenesis do not encompass the entire process of bone formation since there has been no interaction with vascularization and circulating factors and cells, but collectively they represent essential landmarks that can serve in screens for optimization of both therapeutics and treatments prior to evaluation in animal models.

• It is critical to link the in vitro studies with something of physiologic relevance, the idea of which is completely lacking in the manuscript.

Response: We believe that the data and interpretation in this manuscript have been fairly presented and compared (outlined above) to key elements of in vivo de novo bone formation and are thus of physiological relevance. We cannot imagine successful in vivo bone formation in the absence of alkaline phosphatase activity, differentiation of osteocytes, extensive deposition of collagen and mineral, the formation hydroxyapatite, and increase in bone volume and bone mineral density by micro-CT. The model and the data do not predict clinical benefit but provide a platform for further investigation of effective PEMF parameters and mechanisms which may subsequently lead to evaluation of clinical benefit.

• Editors request to justify publication of your paper in PLOS ONE rather than in a specialized biomaterials journal.

Response: The following modified paragraph now ends the Discussion. “PEMF optimization may lead to more effective treatment of a variety of musculoskeletal ailments, in particular those involving bone metabolism [1]. The osteogenic ring culture system is expected to facilitate this endeavor as well as contribute to preclinical screens of biologic and pharmacologic agents targeted at bone deposition, turnover and resorption. Additionally, ring culture may enhance investigation in other experimental systems that benefit from 3D culture of single or multiple cell types, but are limited by the geometry of spheroid or organoid culture such as metastasis, vasculogenesis, and skin equivalents (Lee, Y-S).” 

In addition, it is worth noting that PLOS One has just published a paper (Colombina, A) that assesses the response of tendon cells in monolayer culture to PEMF, and concludes, as we do, that additional optimization of PEMF parameters in culture is warranted before proceeding to animal models and clinical evaluation.

Lee, Yun-Shain, Hsu, Tim, Chiu, Wei-Chih, Sarkozy, Heidi, Kulber, David A, Choi, Aaron, Kim, Elliot W, Benya, Paul D, Tuan, Tai-Lan. Keloid-derived, plasma/fibrin-based skin equivalents generate de novo dermal and epidermal pathology of keloid fibrosis in a mouse model. Wound Repair Regen. 2016 Mar;24(2):302-16. doi: 10.1111/wrr.12397.

Colombini A, Orfei CP, Vincenzi F, de Luca P, Ragni E, Viganò M, et al. A2A adenosine receptors are involved in the reparative response of tendon cells to pulsed electromagnetic fields. PLoS One. 2020;15: 1–14. doi:10.1371/journal.pone.0239807

6. PLOS authors have the option to publish the peer review history of their article (what does this mean?). If published, this will include your full peer review and any attached files.

Do you want your identity to be public for this peer review? For information about this choice, including consent withdrawal, please see our Privacy Policy. 

Reviewer #1: No

Reviewer #2: No

The Preflight Analysis and Conversion Engine (PACE)

Response: The PACE digital diagnostic tool, https://pacev2.apexcovantage.com/ is currently being used to test the submitted figures and is delayed due to a consistent unresolved error message. We realize that this may delay publication if the manuscript is accepted. We will notify PLOS One immediately when the figures have passed this test.

---

## [Editor Report · Decision Letter 1]

7 Dec 2020

Pulsed electromagnetic field (PEMF) transiently stimulates the rate of mineralization in a 3-dimensional ring culture model of osteogenesis

PONE-D-20-27626R1

Dear Dr. BILLI,

We’re pleased to inform you that your manuscript has been judged scientifically suitable for publication and will be formally accepted for publication once it meets all outstanding technical requirements.

Kind regards,

Gary Stein

Academic Editor

PLOS ONE

Additional Editor Comments (optional):

The authors have adequately responded to the critiques. The revised manuscript is recommended for publication.
---

## [Editor Report · Acceptance letter]

2 Jan 2021

PONE-D-20-27626R1 

Pulsed electromagnetic field (PEMF) transiently stimulates the rate of mineralization in a 3-dimensional ring culture model of osteogenesis 

Dear Dr. Billi:

I'm pleased to inform you that your manuscript has been deemed suitable for publication in PLOS ONE. Congratulations! Your manuscript is now with our production department. 

Kind regards, 

on behalf of

Dr. Gary Stein 

Academic Editor

PLOS ONE